# Temporal stability of long-term satellite and reanalysis products to monitor snow cover trends

Ruben Urraca[1,*] and Nadine Gobron[1]

[1]European Commission, Joint Research Centre, Via Fermi 2749, I-21027 Ispra, Italy

**Correspondence:** Ruben Urraca (ruben.urraca-valle@ec.europa.eu)

**Abstract.**

Monitoring snow cover to infer climate change impacts is now feasible using Earth Observation data together with reanalysis products (derived from earth system model and data assimilation). Temporal stability becomes essential when these products are used to monitor snow cover changes over time. The stability of satellite products can be altered when multiple sensors are combined into a single product and due to the degradation and orbital drifts in each individual sensor. The stability of reanalysis datasets can be compromised when new observations are assimilated into the model. This study evaluates the stability of some longest satellite-based and reanalysis products (ERA5, 1950-2020, ERA5-Land, 1950-2020, and NOAA CDR, 1966-2020) by using 527 ground stations as reference data (1950-2020). Stability is assessed with the time series of the annual bias in snow depth and snow cover duration of the products at the different stations.

Reanalysis datasets face a trade-off between accuracy and stability when assimilating new data to improve their estimations. The assimilation of new observations in ERA5 improved significantly its accuracy during the recent years (2005-2020) but introduced three negative step discontinuities in 1977-80, 1991-92, 2004-05. By contrast, ERA5-Land is more stable because it does not assimilate directly snow observations, but this leads to a worse accuracy despite having a finer spatial resolution. NOAA CDR shows a positive artificial trend since 1990-1995 in fall and winter that could be related with the increasing number of satellite data used. The magnitude of most of these artificial trends/discontinuities is larger than actual snow cover trends and Global Climate Observing System (GCOS) stability requirements. Using these products in seasons and regions where artificial trends/discontinuities appear should be avoided.

The study also updates snow trends (1955-2015) over local sites in the North Hemisphere (NH) corroborating the retreat of snow cover, driven mainly by an earlier melt and recently by a later snow onset. In warmer regions such as Europe, snow cover decrease is coincident by a decreasing snow depth due to less snowfall, while in drier regions such as Russia snow cover retreats despite the increasing snow depth observed.

## 1 Introduction

Ground snow cover plays a very important role in the climate system due to its high albedo, thermal insulation and contribution to soil moisture and runoff, therefore has been defined as an essential climate variable (ECV) by the Global Climate Observing System (GCOS)(GCOS, 2016). A snow cover decrease has been observed globally during the last decades (Stocker et al., 2013;

Blunden and Arndt, 2020), which is at the same time a consequence and a cause of global warming. Snow cover retreat has led to a positive snow-albedo feedback of [0.3, 1.1 $Wm^{-2}K^{-1}$] in the Northern Hemisphere (NH) (Stocker et al., 2013). The increase in global net solar energy flux due to snow cover loss ranges from 0.10–0.22 $Wm^{-2}$ ($\pm 50\%$; medium confidence) depending on dataset and time period (Meredith et al., 2019). The effects are amplified in the poles, particularly over the Arctic, which has warmed at more than twice the global rate during the last 50 years driven by strong snow-albedo feedback. Two pronounced warming peaks during snow onset (October-November) and snow melt (April-May) seasons can be currently observed (Brown et al., 2017). The snow loss is not only affecting the global energy budget but also other systems such as the water cycle, vegetation, soil conditions, global atmospheric circulation, and human activities, among others (Callaghan et al., 2011).

Ground stations provide the most accurate snow measurements, but their spatial representativeness is very limited in mountain regions or places with heterogeneous land cover. Besides, ground measurements are scarce in remote regions where the snow loss effects are greater such as the Arctic and high elevations. Long-term snow measurements are particularly limited in the Southern Hemisphere (SH) (Stocker et al., 2013). Therefore, gridded snow products have become crucial to evaluate globally the snow trends during the last decades (Brown et al., 2017). Existing products report a wide range of snow parameters including snow mass, e.g., snow depth (SD), snow water equivalent (SWE), snow density ($\rho_S$), and snow cover, e.g., binary snow cover (SC), snow cover fraction (SCF), snow cover duration (SCD) and snow cover extent (SCE).

Satellite products estimate snow properties based either on the visible/infrared or the microwave spectral regions (Frei et al., 2012). Estimating snow cover from optical data is straightforward but presents limitations related to cloud cover, vegetation, and non-illuminated regions. Some examples include the National Oceanic and Atmospheric Administration (NOAA) Interactive Multisensor Snow and Ice Mapping System (IMS, 1998-present) (Chiu et al., 2020), the historical NOAA weekly SCE charts (NOAA CDR, 1966-present) (Estilow et al., 2015), National Aeronautics and Space Administration (NASA) Moderate Resolution Imaging Spectroradiometer (MODIS) snow cover products (2000-present) (Hall et al., 2006), the Japan Aerospace Exploration Agency (JAXA) SCE product (GHRM5C, 1979-present) (Hori et al., 2017), which combines AVHRR and MODIS imagery, or the NH SCE 1km product produced by Copernicus Global Land Service (CGLS) in near real-time using Suomin-NPP/VIIRS images (Schwaizer et al., 2020). On the other hand, microwave-based methods exploit the scattering of microwave radiation by snow grains, being able to estimate snow mass parameters such as SWE under all-sky conditions. However, they have coarser resolutions ($\geq$ 25 km), and their uncertainty increases over deep snowpacks (SWE > 150 mm) (Pulliainen et al., 2020). GlobSnow (1979-present) (Luojus et al., 2021) and the snow Climate Change Initiative (snow CCI, 1979-2018) (Solberg et al., 2020), both developed by the European Space Agency (ESA), combine passive microwave retrievals with station observations to estimate SWE and SCE. More detailed reviews can be found at Frei et al. (2012) or the SnowPEX project (https://snowpex.enveo.at), an international joint effort to inter-compare satellite SWE and SCE estimations (ESA, 2020; Mortimer et al., 2020).

Global reanalyses appear as an increasingly appealing option for climate studies due to their long-term global coverage of multiple atmospheric, land and ocean variables. They provide estimations of most snow parameters such as snowfall, snowmelt, snow mass and snow cover. The latest generation of global reanalysis includes ERA5 (1950-present) from the Copernicus

Climate Change Service (C3S) (Albergel et al., 2018), MERRA-2 (1980-present) from NASA (Gelaro et al., 2017) and JRA-55 (1953-present) from the Japanese Meteorological Agency (JMA)(Kobayashi et al., 2015). They mainly differ in their spatial resolution, the complexity of their snow schemes (Krinner et al., 2018), and the amount and type of observations assimilated. However, despite their recent improvements, their accuracy is still constrained by their coarse spatial resolutions (30-60 km)
(Urraca et al., 2018; Mortimer et al., 2020; Orsolini et al., 2019; Bian et al., 2019).

The temporal coverage of satellite products is limited to that of the satellite/sensor used, so different satellite instruments are combined to produce Climate Data Records (CDRs). For instance, JAXA GHRM5 combines optical data from NOAA's AVHRR and MODIS sensors, whereas both ESA GlobSnow and ESA CCI SWE combine passive microwave data from SMMR, SSM/I and SSMIS sensors. The transition periods between different sensors are the main source of instability in these products,
but stability issues can also arise due to sensor degradation and orbital drifts (e.g., AVHRR data). The increasing number of satellite sources can also alter the stability of products derived manually by analysts from multiple sources of satellite imagery (e.g., IMS and NOAA CDR). On the other hand, global reanalyses are generally available since the start of the satellite era or before, but the amount and type of data assimilated changes temporally (Mudryk et al., 2015). All these issues can introduce artificial trends or discontinuities in long-term satellite and reanalyses products. Characterizing their stability is
therefore critical, particularly for climate applications. Stability is defined by GCOS as the extent to which the uncertainty of measurement remains constant with time (GCOS, 2016). GCOS stability requirements for snow cover are 10 mm/decade for SD and SWE, and 4%/decade for SCE. These requirements refer to the maximum acceptable change in systematic error per decade.

The goal of this study is to evaluate the temporal stability of ERA5, ERA5-Land and NOAA CDR, which are some of the
snow products with the longest temporal coverage. Indeed, ERA5 and NOAA CDR are currently the longest-term global 4D-Var reanalysis and satellite CDR, respectively. Particularly, the NOAA CDR has been the one most commonly used for climate studies including the IPCC AR5 (Stocker et al., 2013) or the State of the Climate (Blunden and Arndt, 2020). We also include ERA5-Land as an example of land reanalysis with a finer spatial resolution (9 km) (Muñoz Sabater, 2019). The stability of the products is evaluated against 527 ground stations over the NH measuring snow from 1950 to 2020. The study also updates the
snow cover trends in the Northern Hemisphere during from 1955 to 2015. Snow depth and snow cover trends are evaluated with in-situ data due to the discontinuities and trends found in gridded datasets. Snow cover extent could be only evaluated by inter-comparing the three gridded products.

## 2  Data and Methods

### 2.1  Snow products

#### 2.1.1  ERA5

ERA5 is the latest global climate reanalysis produced by the European Centre for Medium-Range Weather Forecasts (ECMWF) within the Copernicus Climate Change Service (C3S) providing hourly data of atmospheric, land and sea parameters. ERA5

implements a 4D-var assimilation system based on the Integrated Forecast System (IFS) CY41R2, with 137 vertical pressure levels and a spatial resolution of around 31 km. The amount of data assimilated increases temporally, with the first satellite

observations starting in 1979. ERA5 uses the H-TESSEL land surface model which implements a single-layer snow model (Dutra et al., 2010). SCF is a diagnostic variable calculated as $min(1, SD[cm]/10)$. It assimilates both in situ and satellite snow observations. The number of in-situ snow observations assimilated has been progressively increasing since 1979, when the first stations were included, reaching 4689 stations in 2020. Since 2004, ERA5 also assimilates the IMS product but only over altitudes below 1500 m. IMS (Chiu et al., 2020; Orsolini et al., 2019) is produced by NOAA combining microwave, visible

and infrared satellite images, as well as manual analysis input, to produce the binary NH snow cover with a spatial resolution of 24 km (since 1997), 4 km (since 2004) and 1 km (since 2014). The IMS 4 km binary SC is assimilated into the ERA5 model by assigning to all IMS snow-covered pixels a snow depth of 5 cm (SCF = 50%).

ERA5 is currently available from 1979 onward at the C3S Climate Data Store (CDS) (Albergel et al., 2018). A preliminary back extension (1950-1978) was recently released (Bell et al., 2020). In this study, both versions are used making a combined

temporal coverage of 71 years (1950-2020). The snow parameters available at the CDS are SWE and $\rho_s$ at hourly and monthly resolution. In this study, hourly SWE and $\rho_s$ were used to calculate the hourly SD, and then aggregated to obtain the average daily SD.

### 2.1.2 ERA5-Land

ERA5-Land is a replay of the land component of the ERA5 climate reanalysis, forced by meteorological fields from ERA5.

It has been produced with the ERA5 land model H-TESSEL (version IFS CY45R1) without coupling the atmospheric model and without directly assimilating observations to make the simulations computationally affordable (Muñoz Sabater, 2019). This allowed the implementation of some improvements for land surface applications such as a finer spatial resolution of around 9 km. The snow model is the same as in ERA5, but observations are not directly assimilated. Neither in-situ snow depth measurements nor IMS data are directly assimilated by ERA5-Land. ERA5-Land is still influenced indirectly by the

snow observations (and observations of other variables) assimilated by ERA5, because ERA5 atmospheric variables are used to control the simulated ERA5-Land fields, which is known as ERA5 atmospheric forcing.

Similarly to ERA5, ERA5-Land spans from 1950 to the present. The main ERA5-Land dataset goes from 1981-present, and the ERA5-Land back extension covers 1950-1980. Compared to ERA5, ERA5-Land provides SD as a diagnostic parameter in the CDS, besides SWE and $\rho_s$. However, we did not use the diagnostic SD, and we calculated the daily SD from the hourly

SWE and $\rho_s$ to keep consistency with the method applied in ERA5.

**Table 1.** Description of the snow products used in the study.

| Product | Producer | Type | Spatial Coverage | Temporal Coverage | Spatial resolution | Temporal resolution |
|---------|----------|------|------------------|-------------------|--------------------|--------------------|
| ERA5 | ECMWF/C3S | reanalysis | global | 1950-1978, 1979-2020 | 0.25° × 0.25°(∼31 km) | 1 hour |
| ERA5-Land | ECMWF/C3S | land reanalysis | global | 1950-1980, 1981-2020 | 0.1° × 0.1°(∼9 km) | 1 hour |
| NOAA CDR | NOAA/NSIDC | satellite | Northern Hemisphere | 1966-2020 | 720 x 720 pixels (∼25 km) | 1 week |

### 2.1.3 NOAA CDR

The NOAA weekly SCE (NOAA CDR) (Estilow et al., 2015) is the longest satellite CDR currently available and the one most widely used for climate applications. It spans from 4 October 1966 up to the present with only 9 months missing (Jul 1968, Jun-Oct 1969, Jul-Sep 1971). Before June 1999, SCE charts were manually produced by trained NOAA meteorologists based on different sources of visible satellite imagery, and then digitalized into a 89×89 Cartesian grid laid over a NH stereographic projection (∼190.5 km) (Robinson et al., 1993). Since then, weekly charts are based on the daily IMS 24 km binary snow cover (Helfrich et al., 2007). The two methodologies overlapped for two years (June 1997 - May 1999) that were used to minimize the impact of the transition in the CDR. Based on this overlap, the conversion from the daily 24 km IMS product to the weekly (Tuesday to Monday) 190.5 km NOAA CDR was made using the Monday IMS estimates and setting as snow-covered those NOAA CDR pixels where a 42% of IMS pixels indicated snow. Therefore, weekly SCE maps are heavily weighted towards the end of the mapping week (Estilow et al., 2015). In this study, the NH SCE version 4 available at the National Snow & Ice Data Center (NSIDC) is used, which re-grids the original NOAA CDR to a NH EASE-Grid 2.0 projection of 25x25 km (Brodzik and Armstrong, 2013).

### 2.2 In-situ snow measurements

In situ snow daily observations were obtained from the Global Historical Climatology Network (GHCN) managed by NOAA's National Centers for Environmental Information (NCEI), the All-Russia Research Institute of Hydrometeorological Information, World Data Centre (RIHMI-WDC), and the Environment and Climate Change Canada (ECCC).

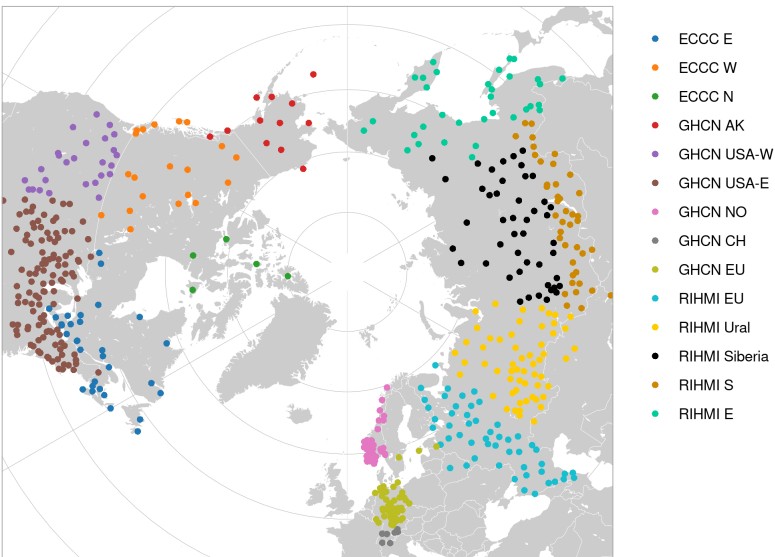

**Figure 1.** Spatial distribution of the stations used in the study.

GHCN is an integrated database with more than 100 000 stations across the globe providing daily measurements of land variables since 1981 (Menne et al., 2012). The stations are divided into 4 main groups based on the data source: (i) US collection, the largest one, (ii) international collection, obtained through personal contacts, (iii) government data exchange, data collected through official GCOS or bilateral agreements, and (iv) the global summary of the day from SYNOP reports. The data can be freely accessed at ftp://ftp.ncdc.noaa.gov/pub/data/ghcn/daily/

RIHMI-WDC contains 620 stations measuring snow since 1882 (Bulygina et al., 2011, 2009). They measure both the snow depth at the station and the snow cover fraction in the surrounding region, which is estimated visually every morning. Snow course surveys are also available every 5 or 10 days depending on the season but are not used in the present study. The dataset includes an automatic quality control procedure that flags potentially erroneous snow depth measurements. All values flagged by this procedure were discarded. The data can be freely accessed at http://meteo.ru/english/climate/snow1.php

ECCC is the primary in-situ network for monitoring snow cover trends in Canada (Brown et al., 2021). ECCC network is composed by manual and automated stations. Manual stations based on ruler measurements started in 1883, but global coverage of Canada was not achieved until mid-1950s. The manual network peaked around 1980s with more than 1600 stations measuring daily snow depth, but it has been declining since 1990s due to the closure of stations and curtailment of manual SD-observing programs. Automated stations with Campbell Scientific SR50 or SR50A sonic snow depth sensors have been replacing manual ones since 1990s, but these efforts have not compensated the high number of manual stations closed. Only manual stations fulfilled the selection procedure for this study. As suggested by Brown et al. (2021), gap-filled SD values within 14 days of an observation were used in this study.

All stations measuring daily SD from each network between 1950 and 2020 were used. Over Canada, this period was reduced to 1955-2015 due to high decrease of ECCC stations before 1955 and after 2015. Only stations with more than 10 snow days per year and 90% of valid years in the study period were exploited. Missing values are frequent in snow measurements due to the practice of only recording days with snow presence (Pirazzini et al., 2018). Filling them systematically with zeros would introduce a negative bias in the measurements since some missing values could be truly missing. To avoid this, only years with less than 5% of missing days were used. In some stations, missing values had already been filled with zeros. These cases were identified by flagging years without snow in stations with more than 40 snow days/year. Flagged years were removed after visually inspecting their time series.

Based on the previous methodology, a reference dataset of 527 stations (228 RIHMI, 242 GHCN, 57 ECCC) was created (Fig. 1). Out of them, 235 are currently assimilated by ERA5. These stations were kept for the stability analysis, since their addition to ERA5 may explain some of the discontinuities observed, but were removed from the accuracy analysis to guarantee the independence of the validation set. The validation set was manually divided into the following spatial regions based on the snow patterns and the performance of the snow products: ECCC N (North Canada, 5), ECCC E (Eastern Canada, 31 stations), ECCC W (Western Canada, 21 stations), GCHN AK (Alaska, 12 stations), GHCN USA-W (Western USA, 26 stations), GHCN USA-E (Eastern USA, 116 stations), GHCN EU (Central Europe, 43 stations), GHCN CH (Switzerland, 7 stations), GHCN NO (Norway, 38 stations), RIHMI EU (European Russia, 49 stations), RIHMI Ural (Ural region, 59 stations), RIHMI Siberia (46 stations), RIMHI S (Southern Siberia, 41 stations), RIMHI E (Eastern Russia, 33 stations).

## 2.3 Spatial representativeness of in-situ snow observations

The spatial representativeness of in-situ observations is critical to conduct point-to-pixel validations, particularly when evaluating coarse products such as global reanalyses. The extent to which point observations are representative of their larger surrounding depends on the geophysical variable and the characteristics of the surrounding terrain, among other factors. The spatial representativeness of in-situ observations was assessed based on the method proposed by Schwarz et al. (2017) for downward solar radiation measurements. This method uses a high-resolution product to evaluate the variability of a geophysical variable within the pixels of the coarser product being validated. For that, the high-resolution pixel collocated with the station is compared against the mean of the high-resolution pixels contained by the coarser pixel. The method includes (i) a correlation analysis and (ii) an estimation of the spatial sampling error (SSE), which arises when estimating a variable over a large area (i.e., the coarse pixel) from a point observation.

The high-resolution product used was NOAA's IMS 1km, which provides daily binary snow cover since 2014 (Chiu et al., 2020). The coarse products evaluated are ERA5 ($0.25° \times 0.25°$) and ERA5-Land ($0.1° \times 0.1°$). The spatial representativeness was evaluated using all daily IMS maps of snow cover during 2015. For each station, the daily snow cover from the IMS pixel collocated with the station ($SC_d^{station}$) and the mean of IMS pixels contained by coarser ERA5/ERA5-Land pixels ($SC_d^{area}$) were extracted. The coefficient of determination ($R^2$) between both variables was calculated. The spatial sampling error was estimated as the mean absolute deviation (MAD) of daily SC:

$$SSE = \frac{1}{N} \sum_{d=1}^{N} |SC_d^{station} - SC_d^{area}| \tag{1}$$

This SSE metric slightly differs from those proposed by Schwarz et al. (2017), who used a more conservative 95 percentile instead of the mean. Additionally, a new metric was calculated and referred as the spatial sampling bias (SSB) to quantify the systematic error introduced:

$$SSB = \frac{1}{N} \sum_{d=1}^{N} (SC_d^{station} - SC_d^{area}) \tag{2}$$

Both SSE and SSB were originally dimensionless because SC is a binary variable. However, both metrics were multiplied by 365 to analyze the results in terms of annual snow cover duration [days/year], which is more easily interpretable. Stations with ($SSE_{ERA5} > 10 \, days/year$) $OR$ ($SSE_{ERA5Land} > 10 \, days/year$) were flagged as low representative and subsequently removed from the validation after visually inspecting their SCD map around the station (Sect. 3.1).

## 2.4 Validation of snow products

SD and SCD estimations from the snow products were validated against the reference ground measurements. For snow depth, daily SD of ERA5 and ERA5-Land were directly compared against the daily SD measurements. For snow cover duration, daily SC had to be calculated first from daily SD values. The NOAA CDR was also added to this second part of the validation.

Besides low spatially representative stations, stations falling within pixels masked as sea/ocean in the different products (ERA5 - 18 stations, ERA5-Land - 13 stations, NOAA CDR - 18 stations) were also removed for the validation. Note that some sites met the spatial representativeness criteria but fell within a pixel masked as sea by the product.

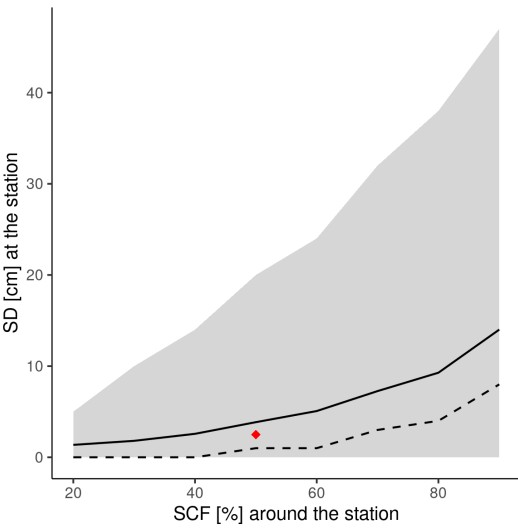

**Figure 2.** Relationship between the snow cover fraction (SCF) around the station and the snow depth (SD) at the station in all RIHMI sites. Solid and dashed lines represent the mean and median value, respectively. The grey shaded region shows the 5 and 95 percentiles. The red dot represents the threshold of 2.5 cm selected to convert SD into SC.

The conversion from snow depth into snow cover is one of the most sensitive aspects when validating snow products. In a previous study performed with the same group of stations by JAXA (Hori et al., 2017), a threshold of 2.5 cm was used to calculate SC based on a local minimum found in the station measurements. The snow cover fraction in the surrounding of the station is visually assessed at RIHMI network (Bulygina et al., 2011). We used these measurements to analyze the correlation between SD at the station and the surrounding SCF (Fig. 2). For a SCF = 50% around the station, the mean and median SD

at the station was 3.95 and 1 cm, respectively. The 2.5 cm threshold used by JAXA falls just in the middle of these values, so we decided to keep the same value in our study. Additionally, we performed a sensitivity analysis to evaluate the impact of threshold used in the snow cover validation metrics (Fig. A2). We analyzed how the SCD bias changes when varying the threshold from 0 to 10 cm by intervals of 2.5 cm. The analysis was made with during 2015 to include the IMS 1km product.

The threshold used to convert SD into SC varies between snow products. Besides, some products provide SCF and others the

binary SC, but the latter is needed to calculate the SCD. For ERA5 and ERA5-Land, all pixels with SCF > 50% (SD > 5 cm) were considered snow-covered. The NOAA CDR already provides the binary SC but at a weekly resolution. The annual SCD was calculated by considering that if a week was flagged as snow-covered, all the days within that week were snow-covered.

### 2.4.1 Validation metrics

The main goal of the study is to assess the stability of long-term snow cover products. For that, the annual mean bias deviation (MBD), hereafter referred simply as the bias, of each product was calculated for SD and SCD values at each station (bias = product - station). Stability was evaluated by analyzing how the annual bias in both SD and SCD changed temporally. Stability was analyzed separately for ECCC, GHCN, and RIHMI stations to discard potential trends or discontinuities in the in-situ measurements, such as major changes in the measuring protocols. If a step discontinuity was found, the difference in the bias between the four years after and before the discontinuity was calculated ($\Delta bias = bias_{after} - bias_{before}$). The relative $\Delta bias$ in % was also calculated as $\Delta bias[\%] = (bias_{after} - bias_{before})/bias_{before}$. The interval of four years was chosen based on the sensitivity analysis of Fig A1. This interval needs to be long enough to remove the effects on inter-annual variations of the snow cover, but too long intervals may be affected by the underlying trends in the bias. Therefore, the shortest interval once $\Delta bias$ has stabilized was chosen. If a trend in the annual bias was found, the decadal trend of the annual bias during that period was computed. The temporal stability of the random error was also analyzed by evaluating how the interquartile range (IQR) of the annual bias changes temporally.

The accuracy of the products was evaluated during those years when the products showed optimal stability: 2005-2020 for ERA-Interim and ERA5-Land. The metrics used were the bias, relative bias, root mean squared deviation (RMSD), and relative RMSD. For SD, the number of daily values below the GCOS accuracy requirements (10 mm) was also calculated. Both accuracy and stability were evaluated annually and seasonally.

## 2.5 Analysis of snow cover trends in the Northern Hemisphere

The trends in SD and SCD were analyzed using the in-situ observations due to the artificial discontinuities and trends found in the snow products. Stations flagged as not spatially representative were kept in this part of the study. The methodology to calculate SCD from SD was the same as that used for the validation. For each variable, decadal trends and annual anomalies for the period 1955-2015 were analyzed. Compared to the stability analysis, the study period was reduced due to the low number of Canadian stations before 1955 and after 2015. The significance of the trends was evaluated with the Mann-Kendall test (Mann, 1945; Kendall, 1975). Note that the density of stations was too low for a complete analysis of NH snow cover trends. Even in regions with good coverage, the heterogeneous density of the stations as well as their different spatial representativeness also prevent the calculation of spatially representative trends. However, our main goal was to estimate the trend magnitude to evaluate the significance of the artificial trends and discontinuities introduced by each product.

The trends in the total NH SCE were analyzed using the three snow products taking into account the stability issues detected during the validation. SCE trends and anomalies were calculated over the period when the three datasets were simultaneously available (1972-2020). In the NOAA CDR, the total NH SCE was calculated by summing the area of all snow-covered pixels. In ERA5 and ERA5-Land, snow-covered pixels were summed taking into account the fraction of the pixel covered by snow.

## 3   Results and discussion

### 3.1   Spatial representativeness of in-situ snow measurements

**Table 2.** Spatial representativeness metrics ($R^2$, SSE - spatial sampling error, SSB - spatial sampling bias) per region in the group of stations selected for the validation after discarding low spatially representative sites.

| | Sites selected (total) | ERA5 grid | | | ERA5-Land grid | | |
|---|---|---|---|---|---|---|---|
| | | $R^2$ | SSE[days/year] | SSB[days/year] | $R^2$ | SSE[days/year] | SSB[days/year] |
| ECCC E | 19(31) | 0.97 | 2.22 | 0.74 | 0.99 | 0.95 | 0.21 |
| ECCC W | 13(21) | 0.99 | 1.16 | -0.39 | 0.99 | 0.77 | -0.15 |
| ECCC N | 2(5) | 0.97 | 1.50 | 1.50 | 0.98 | 1.00 | 1.00 |
| GHCN AK | 5(12) | 0.98 | 1.60 | 1.60 | 1.00 | 0.20 | 0.20 |
| GHCN USA-W | 20(26) | 0.92 | 3.66 | -0.85 | 0.96 | 1.55 | -0.25 |
| GHCN USA-E | 103(116) | 0.96 | 1.78 | -0.17 | 0.98 | 1.02 | 0.01 |
| GHCN NO | 5(38) | 0.94 | 3.01 | -0.60 | 0.95 | 2.01 | -1.60 |
| GHCN CH | 1(7) | 0.94 | 4.01 | -4.01 | 1.00 | 0.00 | 0.00 |
| GHCN EU | 27(43) | 0.92 | 1.63 | -0.89 | 0.95 | 1.00 | -0.26 |
| RIHMI EU | 42(51) | 0.98 | 1.07 | -0.41 | 0.99 | 0.50 | 0.02 |
| RIHMI Ural | 55(57) | 0.99 | 0.69 | -0.00 | 0.99 | 0.55 | -0.22 |
| RIHMI Siberia | 41(46) | 0.99 | 0.98 | 0.10 | 0.99 | 0.76 | -0.46 |
| RIHMI S | 33(41) | 0.98 | 1.67 | -0.82 | 0.99 | 0.58 | -0.03 |
| RIHMI E | 21(33) | 0.99 | 1.15 | -0.86 | 1.00 | 0.43 | 0.14 |
| | 387(527) | | | | | | |

The spatial representativeness criteria were met by 387 out of the 527 snow stations, discarding 140 sites for the validation of gridded products (Table 2, Fig. 3). Stations removed were primarily located in coastal and mountain regions. On the coast, stations overestimate the mean snow cover over the coarse reanalysis pixel because they are located over land while the pixel covers both land and sea (see Fig. 3c). On mountainous regions, the spatial representativeness of the stations decreases due to the large elevation gradients (Fig. 3b). SSE and SSB were very similar in all the stations, which indicates that most of the error introduced by spatial sampling is systematic. Some of these stations showed sampling errors above 100 days/years (>50%). Norway was the region most affected by the station removal, with 87% of its stations discarded due to the combination of an irregular coast surrounded by high mountains. Most of the stations were removed because they did not meet the SSE threshold for the coarser ERA5 grid. However, 17 out of the 104 stations removed passed the threshold for ERA5 grid (0.25°x0.25°) but not for ERA5-Land (0.1°x0.1°). This was the case of sites with high spatial variability of snow cover, where the mean SC over the coarse pixel agreed with the station value just by chance. Implementing the spatial representativeness test at different

spatial resolutions, or taking into account the spatial variability of the geophysical variable, is therefore critical to identify and remove these cases.

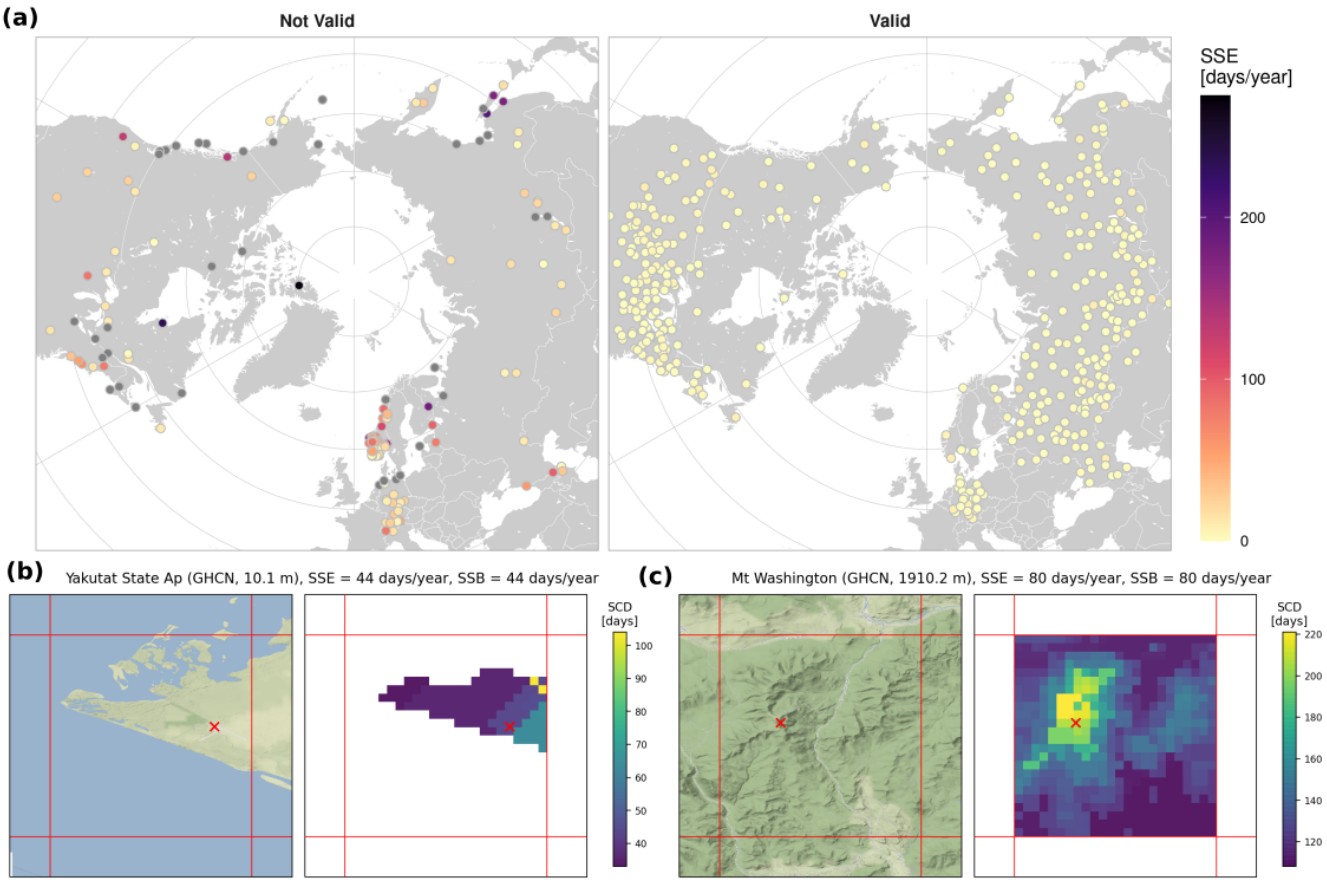

**Figure 3. (a)** Spatial sampling error (SSE) of in-situ snow measurements with respect to ERA5 grid. Spatial distribution of annual snow cover duration (SCD) from IMS 1km in two of the stations flagged as low spatially representative: **(b)** mountain station, **(c)** coastal station. Red lines represent ERA5 grid. The red cross shows the station location. Stations in grey have an IMS snow cover equal to zero. Terrain map tiles by Stamen Design, under CC BY 3.0. Data by OpenStreetMap, under ODbL.

The resulting group of 387 stations used for the validation have an average $R^2 > 0.91$ and $SSE < 4.01\,days$ in all regions. Not representative stations were removed for both accuracy and stability assessment, despite the effects on stability are much lower because SSE and SSB are usually constant in time. The spatial representativeness of the stations was not analyzed for the NOAA CDR grid. Despite the version used in this study has a similar resolution to ERA5, the resolution of the original input data (historical NOAA weekly charts) is much coarser ($\sim$190.5 km) and not appropriate for point-to-pixel comparisons. Therefore, this product was not used in the accuracy assessment and was only kept in the stability evaluation as a reference, i.e., to discard potential artefacts/trends in the stations when evaluating the temporal evolution of the bias.

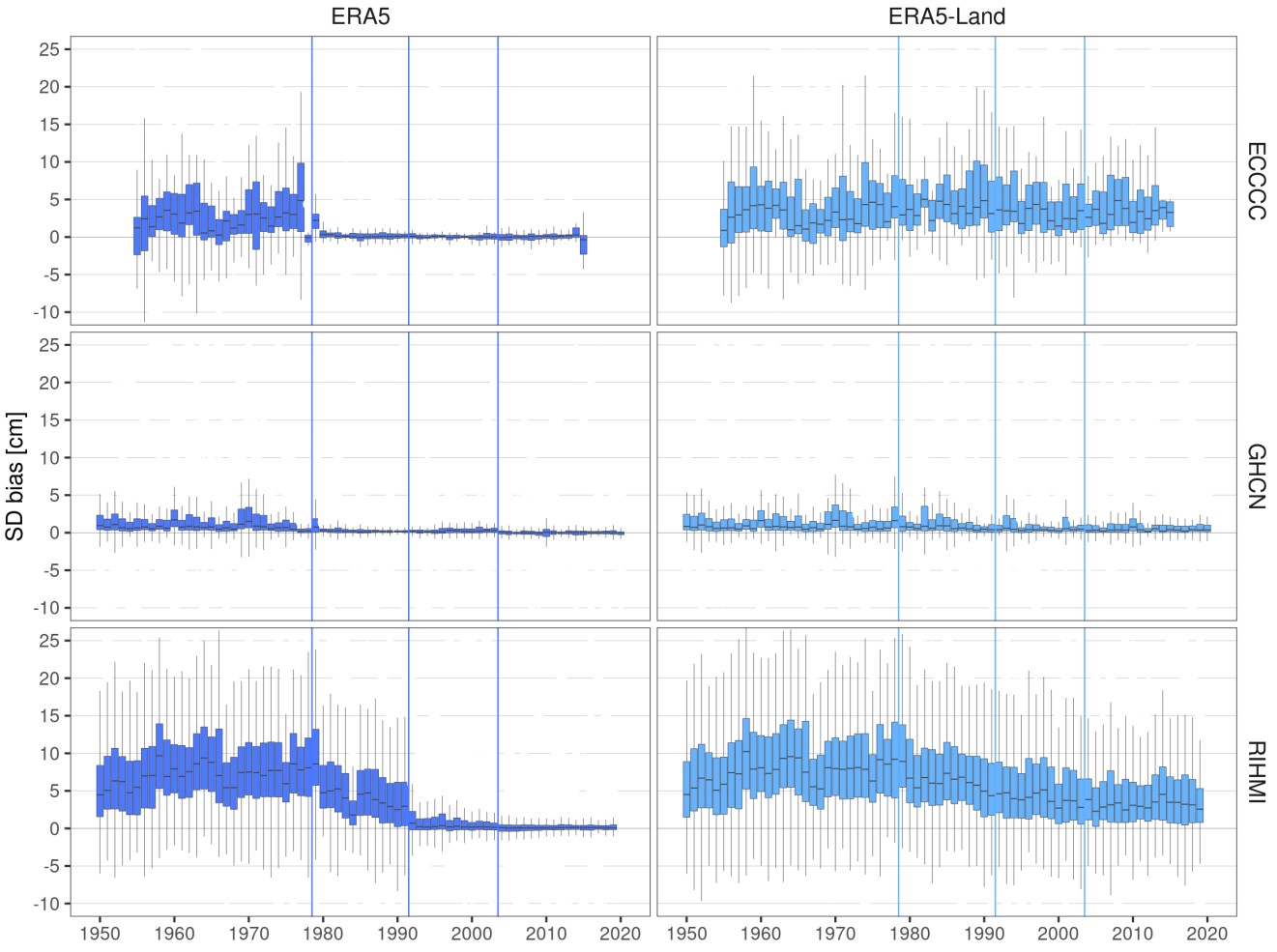

**Figure 4.** Temporal evolution of the annual bias (product - station) in snow depth (SD) per network. Vertical lines show the years when the potential discontinuities in each product occur.

## 3.2 Temporal stability of the products

The analysis of the temporal stability reveals different step discontinuities and trends in the annual bias of both SD and SCD bias (Figs. 4-5) for the three products evaluated. Additional information about the seasonal stability of the bias is available in Figs. A3-A8.

The annual bias of ERA5 significantly decreases in time for both SD and SCD, presenting three negative step discontinuities in 1977-80, 1991-92, and 2004-05, as well as a negative trend between 1980-1991. From 1950 to 2020, the interquartile range (IQR) of the annual bias in SD decreases from [3.5, 11.1 cm] to [-0.2, 0.4 cm] at RIHMI, from [0.3, 1.9 cm] to [-0.2, 0.2 cm] at GHCN, and from [0.1, 5.3 cm] to [-0.3, 0.3 cm] at ECCC. The greater initial bias and greater decrease at RIHMI stations

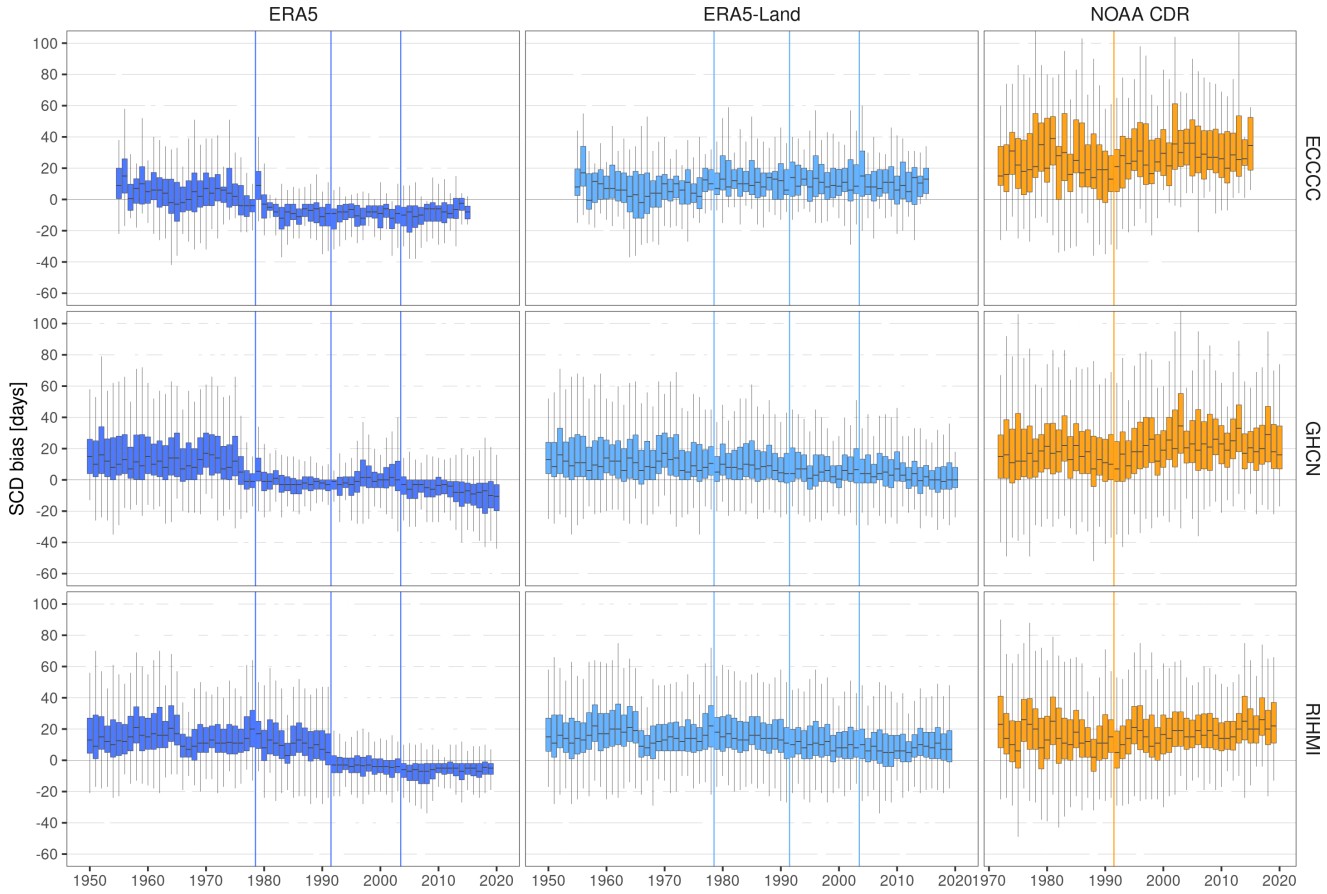

**Figure 5.** Temporal evolution of the annual bias (product - station) in snow cover duration (SCD) per network. Vertical lines show the years when the potential discontinuities/trends in each product occur/start.

are explained by the longer snow season and deeper snowpack over Russia (Bulygina et al., 2011, 2009). The decrease of the
bias is more evident in DJF and MAM seasons (Fig. A4-A5). The magnitude of the bias in SCD is more similar between both networks, with the bias IQR decreasing from [5, 25 days] to [-11, -2 days] at RHIMI, from [2, 26 days] to [-13, -1 days] at GHCN, and from [-5, 15 days] to [-15, -3 days] at ECCC. The negative change of the bias in both SD and SCD is driven by three step-wise discontinuities. They appear in both networks except for the 1992 discontinuity, which is only observed at RHIMI stations. The hypothesis of an artificial discontinuity in RHIMI in-situ measurements was discarded, because the 1992
step discontinuity was not observed in the other products, ERA5-Land and NOAA CDR. Instead, the three discontinuities are more likely caused by the assimilation of new observations by ERA5. In both 1978-80 and 1992 discontinuities, not only the median bias but also the bias variability, a measure of ERA5 random error, was reduced.

ERA5-Land bias also decreases temporally for both SD and SCD but in a more gradual way without showing any step-wise discontinuity. The absence of discontinuities is explained by the lack of direct data assimilation in the ERA5-Land model.

ERA5-Land is still indirectly influenced by observations assimilated in ERA5 through the atmospheric forcing, i.e., the use of ERA5 variables as input to control the simulated ERA5-Land fields. Therefore, the gradual negative trends in ERA5-Land could be indirectly caused through this atmospheric forcing by the three step-wise discontinuities observed in ERA5. Despite being more stable than ERA5, ERA5-Land exhibits always a positive bias but larger bias variability in both SD and SCD. Both the magnitude and variability of ERA5-Land bias are comparable to that of ERA5 before 1980 when no data was being assimilated, whereas ERA5 clearly outperforms ERA5-Land since 1992. Despite having a finer spatial resolution and having being tailored for land surface applications, the quality of ERA5-Land snow estimates is significantly constrained by the lack of data assimilation.

NOAA CDR showed a positive overestimation in SCD and a large bias variability. Both issues were somewhat expected. The positive bias could be explained by changes in the analysis approach to produce the snow charts, which since 1999 considers a pixel snow-covered when only a 42% of the IMS pixels within the pixel were snow-covered. On the other hand, the large bias variability could be also related to the coarse resolution of the original product ($\sim$190.5 km), which makes it inappropriate for point-to-pixel comparisons. This was already stated in the product manual, which recommends using this CDR only for SCE studies over large regions (Estilow et al., 2015). Anyway, as discussed in Sect. 3.1, we kept for the stability assessment because the goal was not to make point SCD estimations but to include a satellite product in the comparison that helps in discarding artificial trends/discontinuities in the in-situ measurements. In this sense, NOAA CDR shows an overall good temporal stability in spring and summer, but a positive trend is observed since 1990 in fall (Fig A6) and winter (Fig A7). The positive trend in fall has been reported previously by several studies (Brown et al., 2017; Brown and Derksen, 2013; Derksen, 2014; Hori et al., 2017) and it is further investigated in Sect. 3.2.2.

### 3.2.1 ERA5 step-wise discontinuities

The magnitude of each ERA5 discontinuity is estimated by calculating the difference in the bias between the four years after and before the discontinuity occurred (Fig. 7, Table A1).

The 1977-80 discontinuity is the most important overall and could be explained by the assimilation of the first satellite products and the first in-situ observations by ERA5 (Fig. 6). As reported in the ERA5 documentation web page (Giusti, 2021), significant improvements were observed in the ERA5 forecast skill after 1978 over regions with scarce conventional observations. Carrying out simulations prior to the satellite era is the main challenge of the ERA5 back extension. The annual $\Delta bias$ is larger in SD (RIHMI = -19.4%, GHCN = -24%, ECCC = -49.8%) than in SCD (RIHMI = -2.5%, GHCN = -7.5%, ECCC = -8.2%). The magnitude of the discontinuity is correlated with the snow depth, having more impact at RIHMI stations and particularly at mountain regions such as the Alps, Southern Russia or Norway. Similarly, the most affected seasons (ranked from largest to smallest) are those with more snow: DJF, MAM, SON. A positive ERA5 bias in SD of similar magnitude was also observed in SD (Orsolini et al., 2019) and SWE (Bian et al., 2019) over the Tibetan Plateau, a region where neither in-situ observations nor the IMS product is assimilated. Orsolini et al. (2019) suggested that the most likely cause was an excessive snowfall precipitation over the Tibetan Plateau, discarding other effects such as snow sublimation due to blowing snow or the SCF threshold. In this study, ERA5 shows also a large positive bias in periods with low data assimilation (before 1980).

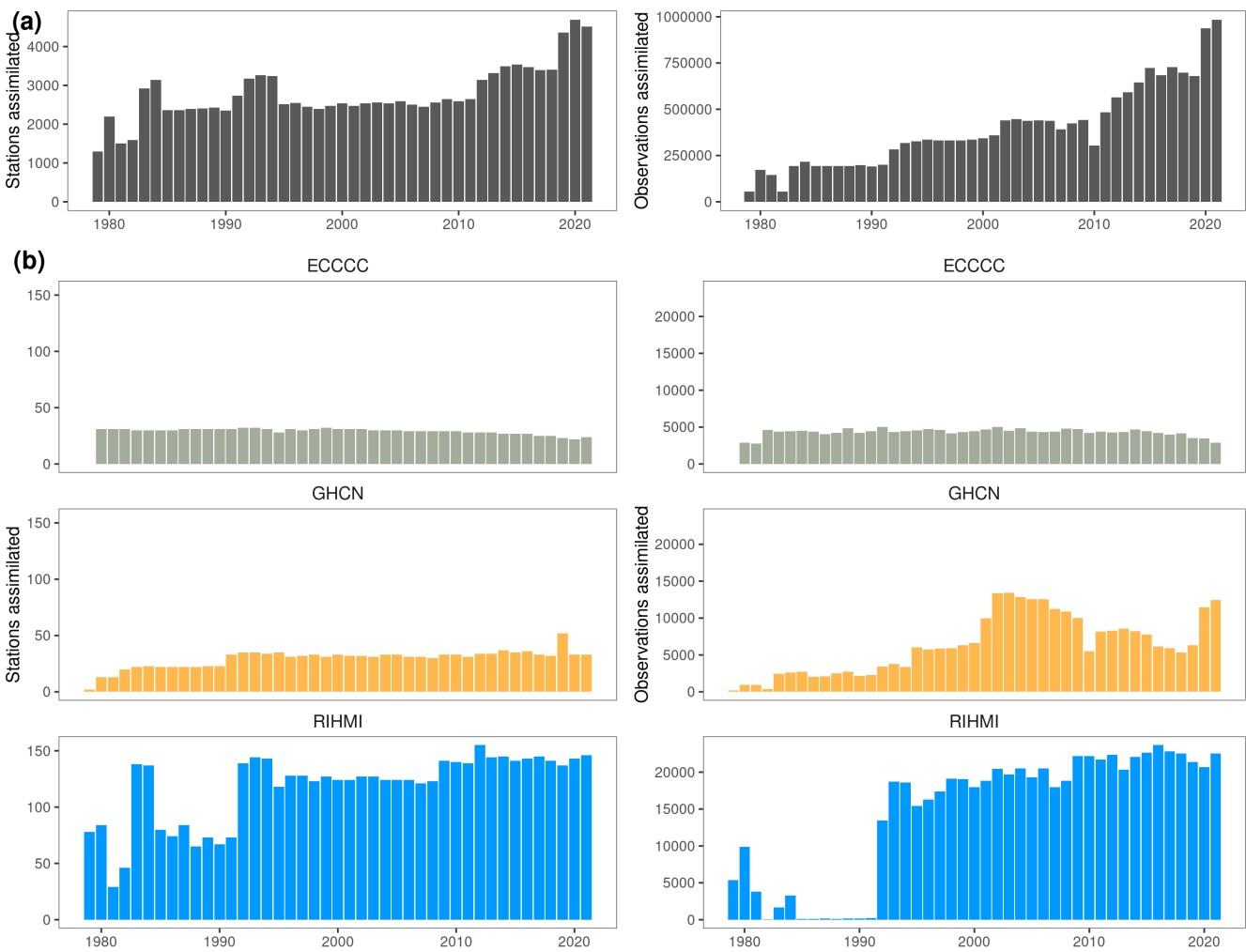

**Figure 6.** Temporal evolution of the in-situ snow stations and observations assimilated by ERA5: (**a**) all stations assimilated by ERA5, (**b**) stations assimilated by ERA5 within our validation set split by network.

Besides, ERA5-Land, which does not directly assimilate observations, also shows a predominantly positive bias in SD. As
suggested by Orsolini et al. (2019), the most likely cause of the snow depth overestimation in both ERA5 and ERA5-Land
could be a precipitation bias, which is only corrected by the assimilation of snow depth observations in ERA5 (after 1979 and
below 1500 m).

The 1991-92 discontinuity presents a similar seasonal pattern in $\Delta bias_{SD}$ than the one in 1977-80, having more impact
(sorted from largest to smallest) in DJF (-52.8%), MAM (-33.1%), and SON (-8.1%), respectively. The main difference is that
the 1991-92 discontinuity is only observed over Eurasia. This step is most likely caused by the assimilation of new in-situ
observations in Russia and China starting from 1992 (Fig. 6). The assimilation of these observations further corrects the large

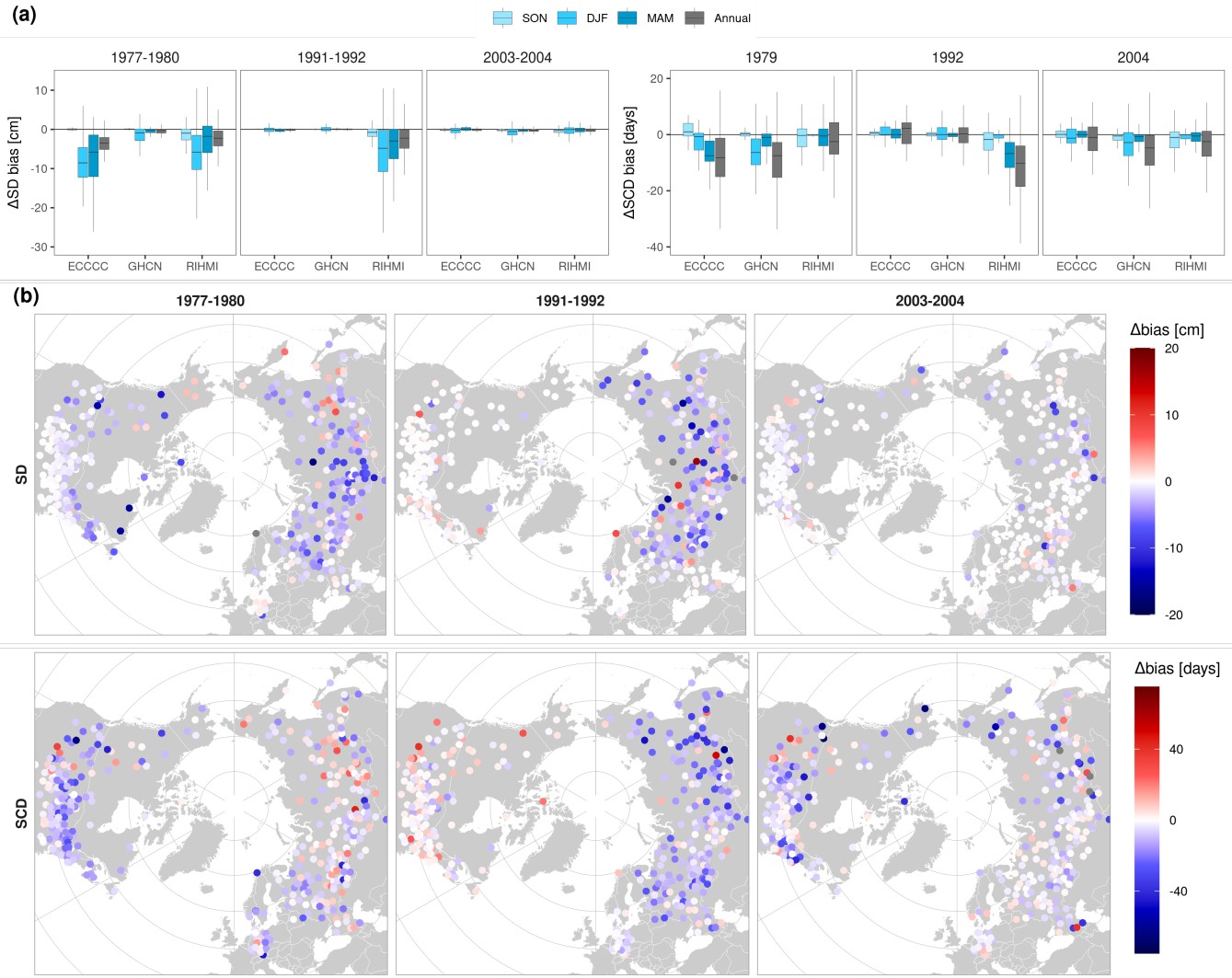

**Figure 7.** Change in the ERA5 bias in snow depth (SD) and snow cover duration (SCD) during 1977-80, 1991-92 and 2004-05 discontinuities. The four years before and after the discontinuity are compared ($\Delta bias = bias_{after} - bias_{before}$). **(a)** Seasonal change in the bias per network. **(b)** Annual change in the bias per station.

positive bias exhibited by most Russian stations is significantly corrected ($\Delta bias_{SD}$ = -18.2 %, $\Delta bias_{SD}$ = -6.3 %), falling within a similar range to that observed over Europe and North America. Similar to the 1980 discontinuity, this discontinuity not only reduced the median bias but also its variability.

The 2004-05 discontinuity, already reported by Mortimer et al. (2020), is caused by the assimilation of the satellite-based IMS product. Compared with the previous ones, this discontinuity has a greater impact on the snow onset-melting detection than on snow depth. The change of the bias is larger in SCD than in SD, and spatially, the discontinuity is larger at GHCN

stations ($\Delta bias_{SD}$ = -17.2%, $\Delta bias_{SCD}$ = -16.1%) than at RIHMI ones ($\Delta bias_{SD}$ = -0.6%, $\Delta bias_{SCD}$ = -1.5%). This could be explained by how the IMS product is integrated into the ERA5 model. Snow-covered pixels are assimilated as 5 cm of snow depth, explaining the relatively low impact in snow depth and the large improvements in the detection of the start/end of the snow season, which are more evident in SON, MAM and regions with low number of snow days. Particularly large changes are also observed in coastal or mountain regions (Rocky mountains, Southern Siberia), which could be related to the benefits of assimilating a product with a finer resolution over these regions, where snow observations assimilated are also scarce. Note that the IMS product was only assimilated below 1500 m, so large improvements observed over mountain regions mostly occur at stations located in the valleys.

Fig.6 shows that significant increases in the number of snow observations assimilated also occurred in 2001-2002 (more European stations), 2011-15 (new stations in Eastern Europe and Scandinavian Peninsula), 2019-20 (new Kazakh stations and more observations in China and Europe). However, discontinuities related to these changes in the assimilated data were observed neither in the SD/SCD time series at the stations nor in the global SCE time series.

Table A1 summarizes the number of stations where ERA5 shows acceptable stability according to the GCOS requirements. This metric was only calculated for SD (stability = 10 mm) because no explicit requirement is made for SCD. The three discontinuities introduced a $\Delta bias_{SD}$ above this threshold in most of the stations, particularly when looking at the winter season alone. Only 10.6%, 12.2%, 58.2% of RIHMI stations, 41.1%, 69.7%, 50.0% of GHCN stations, and 0%, 67.7%, 69.7% of ECCC stations were below the stability threshold in winter during the 1977-80, 1991-92, and 2004-05 discontinuities, respectively. Note that these values would be even larger if looking only at snow-covered days in regions such as USA or Europe where snow does not last the full winter season.

### 3.2.2 NOAA CDR bias trend in fall and winter

NOAA CDR exhibits a positive trend in SCD bias in SON and DJF (Fig. 8, Table S2). The trend steadily starts from 1990-1995 extending until almost the end of the CDR. We quantify this issue by calculating the decadal trend of the bias over the 1992-2015 period. A positive artificial trend in NOAA CDR during SON was already documented by different authors (Hori et al., 2017; Brown and Robinson, 2011; Brown and Derksen, 2013). Brown and Robinson (2011) reported that the positive October SCE trend was an artefact of the NOAA CDR, since this positive trend opposed to several independent snow products and to in situ measurements (Peng et al., 2013). They suggested that the most likely cause could be the increase of satellite data ingested by the NOAA CDR analysts, as well as the increase in the temporal and spatial resolution of these products, which led to a more accurate snow onset detection. In the same line, Mudryk et al. (2017) found that NOAA CDR trends in October and November are non-physical and not consistent with other datasets. Our study corroborates the existence of a significant positive trend around 5-10 days/decade in many Eurasian stations and in some stations of Northern USA. Additionally, our study reveals that the same trend appears in DJF in some European (10 days/decade) and Eastern USA stations (7.2 days/decade), regions where snow onset takes place later than in Russia. If looking at the seasonal SCE anomalies, these artificial trends could explain the SCE recovery observed in SON and DJF exhibited by the NOAA CDR. Fig 5 also corroborates that there is no step discontinuity related to the transition between the two methodologies in 1999, though the positive trend may have been

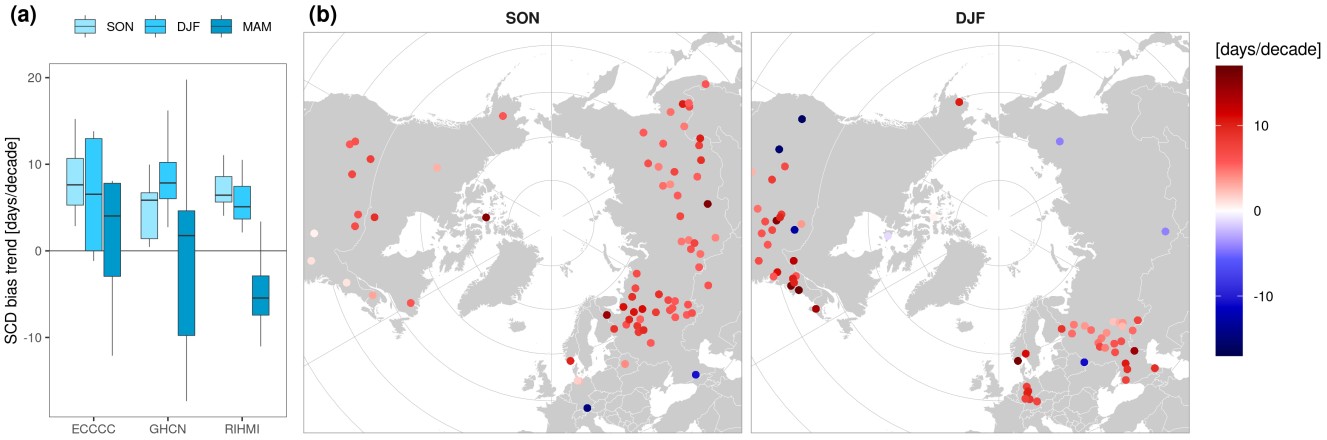

**Figure 8.** Decadal trend of the annual bias in seasonal snow cover duration (SCD) of NOAA CDR from 1992 to 2015 **(a)** per network and **(b)** per station. Only significant trends (p < 0.05, Mann-Kendall) are shown. MAM map was excluded due to the lack of an artificial trend globally during that season.

aggravated after 1999 due to the improved resolution and the increasing number of satellite products ingested by the IMS product.

Brown and Derksen (2013) suggested that the opposite effect during the spring season could be expected but was not
observed. Theoretically, an improved detection of snow melting could lead to a stronger spring trend, introducing an artificial negative trend in the CDR. In this line, Derksen (2014) reported a tendency of NOAA CDR to map less snow in spring since 2007 than the multi-dataset composed by NOAA CDR, MERRA and ERA-Interim. Mudryk et al. (2017) also found that NOAA CDR has a spring trend stronger than other datasets. We analyzed this issue by evaluating the snow cover duration trends in spring. Negative trends in the spring bias only appear in some Russian stations (Fig. 8a). However, the number of
stations showing significant trends in spring is smaller, and the magnitude of these trends is much lower than those in fall and winter. Despite this issue could exist in some specific regions, the impact at global scale is negligible (Fig.A8).

### 3.3 Spatial accuracy of the reanalyses after the last ERA5 discontinuity

Figure 9 shows the performance of ERA5 and ERA5-Land after the last ERA5 discontinuity (2005-2020) to compare both products under the current data assimilation scheme of ERA5. ERA5 estimations are mostly unbiased for SD, with the annual
IQR bias within [-0.1, 0.1 cm] in most regions. Large positive biases only remain over the mountains (Rocky mountains, Southern Russia ranges), which could be related to the lack of IMS assimilation above 1500 m. On the contrary, ERA5-Land constantly overestimates SD in most regions, with the bias IQR within [0.8, 2.9 cm]. Despite its finer resolution, the product quality still degrades in the mountains. Regarding the absolute error, ERA5 shows a RMSE below 1.5 cm in most stations that increases up to 12 cm in mountain stations. On average, 82.6% of daily ERA5 snow depth values meet the GCOS accuracy
requirements, while this number decreases to a 10.5% for ERA5-Land.

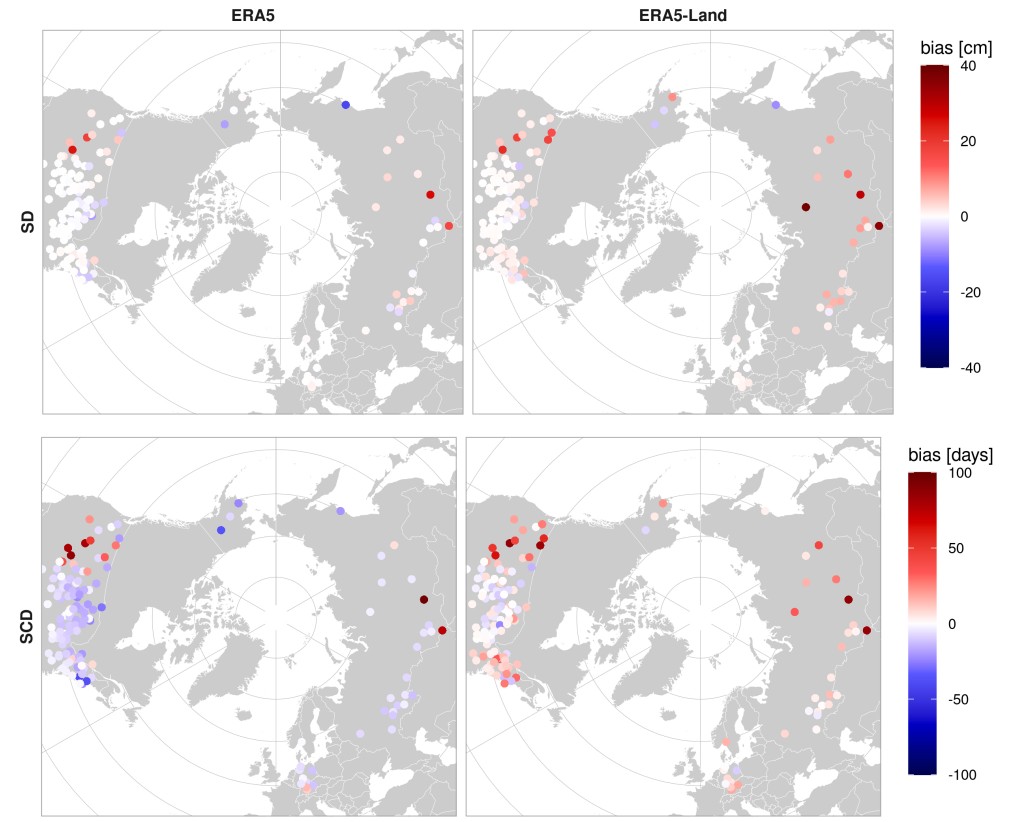

**Figure 9.** Bias (product - station) in snow depth (SD) and snow cover duration (SCD) after the last ERA5 discontinuity (2005-2020). Stations assimilated by ERA5 have been excluded.

In SCD, ERA5 presents a constant underestimation (IQR) of around [-9.4, -5.5 days] while ERA5-Land keeps overestimating [2.4, 11.2 days]. As above mentioned, the SCD bias strongly depends on the threshold used to convert SD to SC. Both ERA5 and ERA5-land use a threshold (5 cm) larger than the one applied to the stations (2.5 cm). This could explain why ERA5 has a negative SCD bias despite having an unbiased snow depth. Indeed, when the ERA5 threshold is applied to the stations (Fig. A2), ERA5 SCD bias is close to zero in the three networks. We could be tempted to use the same threshold in stations and product. However, the thresholds applied by products need to be validated as well, and we can only do it deriving independent thresholds for the station measurements. In this study we have used RHIMI visual observations of snow cover in the station, but other data sources such as high-resolution satellite imagery could be also useful.

We investigated further this issue with a sensitivity analysis that evaluates how the SCD bias changes with different snow depth to snow cover thresholds (Fig. A2). The magnitude of SCD bias is similar between networks, suggesting a good consistency between their measuring protocols. However, the magnitude of SCD bias strongly varies between products. When a threshold of 2.5 cm is used, the mean SCD bias varies as follows: 24.8 days (NOAA CDR), 14.3 days (IMS), 8.0 days (ERA5-

Land) and -6.7 days (ERA5). These differences are the result of the different thresholds applied by the products, as well as their different snow depth biases (in case of reanalysis). Orsolini et al. (2019) already pointed out that the different thresholds applied by reanalysis datasets was one of the main limitations for inter-comparing them. The sensitivity analysis also shows that changing 1 cm the station threshold leads to changes in the annual SCD bias of around 2-3 days. These changes are constant between products but vary between networks (ECCC = 2.8-4.3 days/cm, GHCN = 1.8-2.1 days/cm, RIHMI = 2.6-3.2 days/cm), due to the different snow conditions in each station. Stations with more daily SD values close to the threshold are more affected by changes in the threshold.

## 3.4 Snow cover trends in the Northern Hemisphere

Linear decadal trends in SD and SCD were calculated annually and seasonally over the period 1955-2015 using data from the ground stations (Fig. 10, Table 3). The temporal representativeness of the linear trends was further analyzed by plotting the temporal evolution of the anomalies per spatial region (Fig. A9-A10).

SD trends show large spatial variability. Significantly negative trends are observed over Europe (Norway = -0.9 cm/decade, Central Europe = -0.1, cm/decade) driven by a strong decrease of winter SD, particularly between 1980-1990. On the contrary, significantly positive SD trends are observed over most of Russia, specifically over the Ural Region (+0.9 cm/decade), Siberia (+1.3 cm/decade), and the Sea of Okhotsk (+1.7 cm/decade) driven by a strong increase of both winter and spring snow depth. These trends agree with those reported by Brown et al. (2017) for the Russian Arctic over the 1966-2014 period (SDmax, +0.7 cm/decade). However, as mentioned by Brown et al. (2017), a tipping point is observed around 2000 that reverses the SD increase during the latest years in some Russian regions (e.g., European Russia, Ural region). Negative trends in SD are also observed in Eastern USA and in most of Canada (-0.9 to -1.6 cm/decade)

SCD trends are more spatially homogeneous. A predominantly negative SCD trend is observed globally of around [-2, -4 days/decade] driven by a strong negative trend during the melting season. Largest reductions in annual SCD are observed over Europe (Norway = -6 days/decade, Central Europe = -2.9 days/decade, European Russia = -3.8 days/decade). In Russia, most regions experience a decrease in annual SCD despite their positive SD trends (Siberia = -2.3 days/decade, Southern Siberia = -2.2 days/decade). Only a few stations in the Ural region and Sea of Okhotsk show a longer snow season during the last 70 years. Recalculating the trends for a more recent period 1981-2020 evidences an acceleration of the SCD decrease as well as an increasing weight of a later snow onset in the annual SCD trends. Again, few Eastern USA stations show significant trends. The low number of significant trends compared to that reported by Knowles (2015) could be explained by a recent recovery in winter and spring SCD since 2000-2010 (Fig. A10). Still, the few significant trends observed in USA are predominantly negative with some exceptions around the Great Lakes that Knowles (2015) attributed to an increased precipitation pattern. SCD trends are also consistently negative in most of Canada (-1.5 to -5.3 days/decade) driven by negative DJF trends in coastal regions and negative MAM trends in inland and polar regions.

The large spatial variability in SD trends is explained by the non-linear interactions between temperature and precipitation (Brown and Robinson, 2011). At high latitude, increasing temperatures lead to increasing precipitation due to a moister climate (Thackeray et al., 2019), but snowfall depends on the precipitation phase as well. In relatively warmer climates and maritime

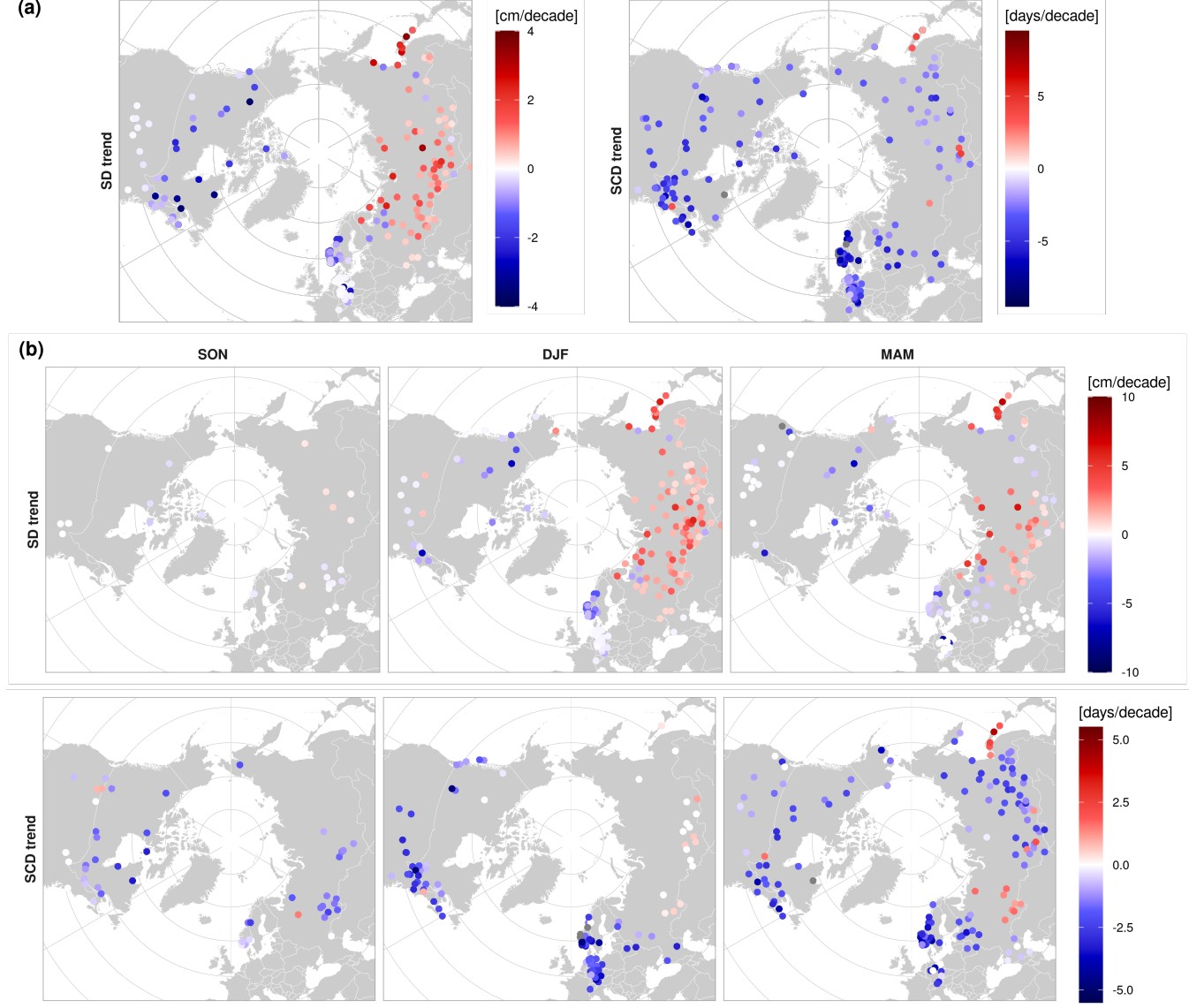

**Figure 10.** (**a**) Annual and (**b**) seasonal decadal trends in snow depth (SD) and snow cover duration (SCD) from 1955 to 2015 based on in-situ measurements. Only statistically significant trends (p-value < 0.05, Mann-Kendall) are shown.

regions (e.g., Central Europe, Scandinavia, European Russia), negative SD trends could be related to a shift in the form of precipitation towards a rainfall-dominated winter (Luomaranta et al., 2019). On the contrary, in colder and drier climates such as Siberia, snow accumulation is limited by moisture availability (Kunkel et al., 2016), so the positive SD trends may be due to warmer and moister weather (Bulygina et al., 2009), and/or to more extreme snow events, which are more likely to occur below the freezing point (Kunkel et al., 2016). Despite these heterogeneous SD trends, SCD trends are consistently negative

globally. SCD reductions over the period 1955-2015 are mainly driven by an earlier melt that is strongly correlated with the increasing spring temperatures amplified by the snow-albedo feedback (Brown et al., 2017; Luomaranta et al., 2019; Bulygina et al., 2009; Matiu et al., 2021). In regions such as Europe, both SD and SCD are decreasing, with the trend towards shallow

snow depth amplifying the shorter snow season. In Russia, spring SCD is also decreasing despite the positive trends in SD. This means that the spring melt driven by warming temperatures overrides any increase in snow accumulation during winter.

Larger variability has been reported for SCD trends during the snow onset season (Brown et al., 2017). However, as also recently suggested by Mudryk et al. (2020), this study evidences the increasing importance of negative SON trends in regions such as Europe, Russia, and the Rocky Mountains, where they have a higher impact than spring trends during the latest years.

**Table 3.** Annual and seasonal decadal trends (median with its 95% CI) in SD [cm/decade] and SCD [days/decade] from 1955 to 2015. Only statistically significant trends (p-value < 0.05, Mann-Kendall) are shown. $N$ depicts the number of stations with significant trends in each region. The median was not calculated in regions that have less than five stations with significant trends.

| | | Annual | | SON | | DJF | | MAM | |
|---|---|---|---|---|---|---|---|---|---|
| | | Trend | N | Trend | N | Trend | N | Trend | N |
| | ECCC E | -1.6 [-3.2, -0.7] | 12 (38.7%) | - | - | - | - | - | - |
| | ECCC W | -0.9 [-1.8, 0] | 10 (50%) | - | 1 (5%) | -1.3 [-4, -0.1] | 9 (45%) | - | 4 (20%) |
| | ECCC N | - | 3 (60%) | - | 3 (60%) | - | 3 (60%) | - | 3 (60%) |
| | GHCN AK | - | 2 (16.7%) | - | - | - | 4 (33.3%) | - | 2 (16.7%) |
| | GHCN USA-W | - | 1 (3.8%) | - | 1 (3.8%) | - | - | -0.1 [-4.5, 0] | 11 (42.3%) |
| SD | GHCN USA-E | -0.2 [-0.5, -0.1] | 15 (12.9%) | - | 5 (4.3%) | -0.4 [-1.6, -0.1] | 14 (12.1%) | -0.1 [-0.3, 0] | 10 (8.6%) |
| | GHCN NO | -0.9 [-1.2, -0.7] | 17 (45.9%) | 0 [-0.2, 0] | 10 (27%) | -2.7 [-3.4, -1.7] | 19 (51.4%) | -1 [-4, -0.6] | 11 (29.7%) |
| | GHCN CH | - | 2 (28.6%) | - | - | - | 2 (28.6%) | - | 2 (28.6%) |
| | GHCN EU | -0.1 [-0.1, 0] | 25 (58.1%) | - | 2 (4.7%) | -0.2 [-0.2, -0.2] | 22 (51.2%) | 0 [0, 0] | 16 (37.2%) |
| | RIHMI EU | 0.2 [-0.8, 0.9] | 13 (25.5%) | - | 4 (7.8%) | 1 [-0.2, 1.9] | 14 (27.5%) | 0 [-1, 0] | 19 (37.3%) |
| | RIHMI Ural | 0.9 [0.7, 1.2] | 26 (45.6%) | -0.2 [-0.6, -0.1] | 7 (12.3%) | 2.2 [1.9, 2.8] | 30 (52.6%) | 1.4 [1.1, 2.1] | 22 (38.6%) |
| | RIHMI Siberia | 1.3 [0.7, 1.7] | 16 (34.8%) | - | 3 (6.5%) | 1.8 [1.3, 3.3] | 22 (47.8%) | 2.4 [1.5, 3.2] | 12 (26.1%) |
| | RIHMI S | 0.4 [0.2, 0.8] | 16 (39%) | - | 1 (2.4%) | 1.2 [0.8, 1.9] | 23 (56.1%) | -0.5 [-1, 0.8] | 11 (26.8%) |
| | RIHMI E | 1.7 [0.3, 2.7] | 11 (33.3%) | - | 1 (3%) | 3.2 [1.1, 5.2] | 11 (33.3%) | 3.4 [0.5, 5.3] | 9 (27.3%) |
| | ECCC E | -4.7 [-5.3, -4.1] | 22 (71%) | -1.3 [-1.8, -0.6] | 14 (45.2%) | -1.8 [-2.7, -0.7] | 14 (45.2%) | -2.5 [-3.1, -2] | 14 (45.2%) |
| | ECCC W | -2.5 [-4.2, -1.5] | 14 (70%) | - | 3 (15%) | -1.2 [-2, -0.2] | 9 (45%) | -2 [-2.4, 0] | 7 (35%) |
| | ECCC N | - | 5 (100%) | - | 2 (40%) | - | - | - | 1 (20%) |
| | GHCN AK | - | 2 (16.7%) | - | - | - | - | - | 3 (25%) |
| | GHCN USA-W | - | 1 (3.8%) | - | 5 (19.2%) | - | - | -1 [-2.9, 0] | 7 (26.9%) |
| SCD | GHCN USA-E | -3.5 [-3.8, -2.5] | 17 (14.7%) | - | 3 (2.6%) | -2.4 [-2.9, -2.1] | 14 (12.1%) | -0.8 [-4, 1.6] | 8 (6.9%) |
| | GHCN NO | -6 [-6.4, -5.4] | 26 (70.3%) | -0.5 [-1.5, -0.2] | 10 (27%) | -3.3 [-4.8, -2.4] | 21 (56.8%) | -2.9 [-3.5, -2.4] | 21 (56.8%) |
| | GHCN CH | - | 3 (42.9%) | - | - | - | 2 (28.6%) | - | 2 (28.6%) |
| | GHCN EU | -2.9 [-3.3, -2.3] | 27 (62.8%) | - | - | -2.1 [-2.6, -1.9] | 23 (53.5%) | -0.2 [-2.7, 0] | 12 (27.9%) |
| | RIHMI EU | -3.8 [-4.8, -2.6] | 13 (25.5%) | - | 2 (3.9%) | -2 [-3.2, -0.8] | 8 (15.7%) | -1.7 [-2.2, -0.6] | 15 (29.4%) |
| | RIHMI Ural | - | 2 (3.5%) | -1.6 [-1.9, -1.5] | 9 (15.8%) | - | 5 (8.8%) | 1.4 [1.1, 1.7] | 9 (15.8%) |
| | RIHMI Siberia | -2.3 [-3.6, -1.7] | 13 (28.3%) | - | 4 (8.7%) | - | 3 (6.5%) | -2 [-2.4, -1.2] | 17 (37%) |
| | RIHMI S | -2.2 [-2.8, -1.7] | 12 (29.3%) | - | 1 (2.4%) | 0 [0, 1.1] | 8 (19.5%) | -1.4 [-2.2, -1.2] | 22 (53.7%) |
| | RIHMI E | -1.6 [-3.4, 4.3] | 8 (24.2%) | - | 1 (3%) | 0 [0, 0.3] | 7 (21.2%) | -1.3 [-2.4, 1.9] | 14 (42.4%) |

The Northern Hemisphere presents an average annual SCE of 23.9 $million\,km^2$ (NOAA CDR) over the 1972-2020 period (common period between the three products). The three products show an annual decrease in NH SCE (Fig. 11), though the SCE trends should be interpreted cautiously taking into account the discontinuities/trends discussed in Sect. 3.2. NOAA CDR is the product typically used for assessing the NH SCE trends. It shows the smallest trend (1972-2020) in annual SCE overall (-0.15 $million\,km^2/decade$, -0.63 $\%/decade$), which is driven by a significant decrease in MAM (-0.61 $million\,km^2/decade$, -2.13 $\%/decade$) and JJA (-0.71 $million\,km^2/decade$, -14.2 $\%/decade$). These seasons are when most snow melts, and again, these reductions are strongly related to the increasing temperature and the snow-albedo amplification. The small decrease in annual SCE compared to that in MAM and JJA is explained by the SCE positive trends of +0.62 and +0.19 $million\,km^2/decade$ in SON and DJF, respectively. This is due to the artificial positive trend in SON and DJF SCD described in Sect. 3.2.2. In this sense, Derksen (2014) estimated that the artificial trend in October SCE for this product could be around 1 $million\,km^2/decade$, which would revert the sign of the SON trend. Other snow satellite products such as JAXA's GHRM5C (Hori et al., 2017) have also reported negative trends of -0.94 and -0.39 $million\,km^2/decade$ for SON and DJF, respectively, during 1980-2020. Negative trends have been observed as well in SON for the group of stations used in this study. All of this corroborates the underestimation of the snow cover retreat in the fall and winter seasons by the NOAA CDR product.

ERA5-Land had better stability than ERA5, showing a small negative trend caused by the ERA5 atmospheric forcing. This is somewhat corroborated in terms of SCE, with ERA5-Land showing just a slightly smaller trend in MAM than NOAA CDR (-0.54 vs -0.61 $million\,km^2/decade$). On the contrary, ERA5 strongly overestimates the SCE decrease throughout all seasons, showing the largest negative trend in annual SCE (-1.07 $million\,km^2/decade$). Among the three ERA5 discontinuities detected, the assimilation in 2004 of IMS snow product has the largest impact overall, leading to large step discontinuity of around -13% (annual SCE) and -30% (SON SCE) in just one year (Fig. 11b). As above discussed, the large impact of IMS on the onset and melting period was explained by how this product is assimilated by the model. Overall, ERA5 should be avoided to analyze the NH SCE trends before 2004. ERA5-Land has better stability but still overestimates the actual snow cover retreat.

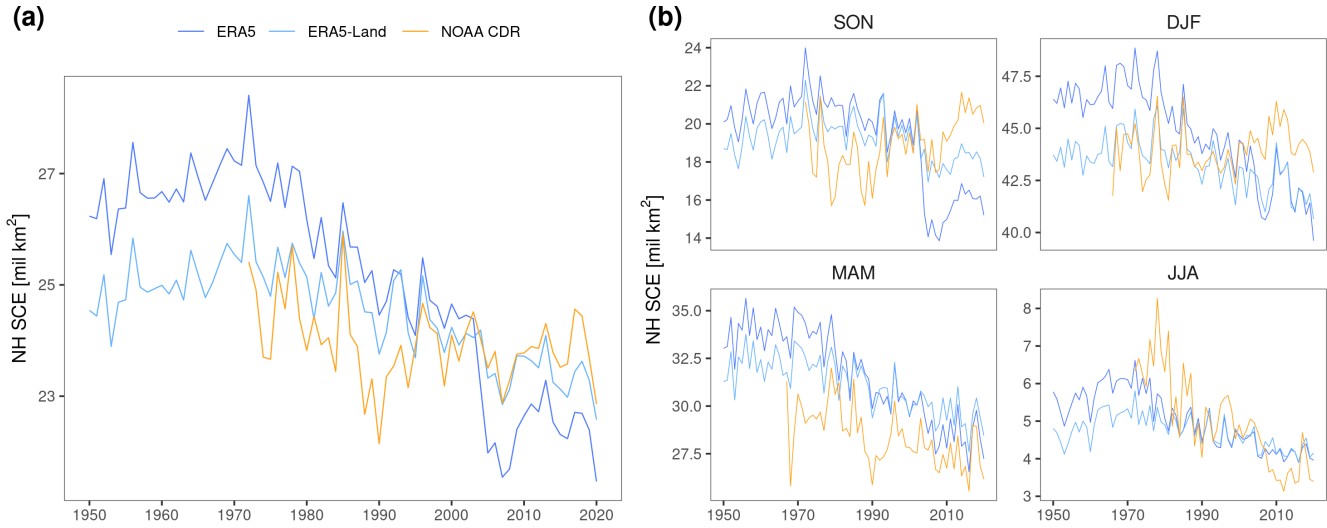

**Figure 11.** Annual (**a**) and seasonal (**b**) NH snow cover extent (SCE) [million $km^2$].

## 4  Stability of the products for snow trend analysis

Global reanalyses appear as an increasingly appealing option for climate studies due to their long-term global coverage of multiple atmospheric, land and ocean variables. Great efforts have been made lately to extend backward global 4D-Var reanalyses with the release of ERA5 back extension (1950-present) and JRA-55 (1958-present). The core of reanalysis products is the data assimilation system that allows combining Numerical Weather Prediction (NWP) simulations with in-situ observations and satellite products. The number of observations available has increased exponentially during the latest years, improving the accuracy of reanalysis estimations and bringing them closer to satellite-based products. However, assimilating new observations creates a trade-off between accuracy and stability. For applications requiring high accuracy such as NWP initialization, more weight is given to new observations assimilated in order to provide the best possible estimations. However, this can introduce temporal inconsistencies in the data records, as observed in ERA5. These challenges increase even more when trying to extend backward reanalysis before the satellite era.

In the case of snow, the present study reveals the high dependence of the ERA5 accuracy on the snow observations assimilated. After 2004, when ERA5 assimilates the IMS snow product and more than 4000 snow stations, it clearly outperforms ERA5-Land, a specific land reanalysis with a much finer spatial resolution (9 vs. 31 km) but without direct assimilation of observations. The strong dependence of the bias on the observations assimilated created a significant negative trend in ERA5 far larger than the 10 mm stability limit of GCOS, particularly in winter. Therefore, the use of ERA5 snow parameters for climate studies before 2004 should be avoided as it artificially overestimates the decrease of all snow-related parameters (SD, SC, SCE). Correcting the systematic bias may be possible (Mortimer et al., 2020) and highly recommendable if using ERA5 before 2004. However, the study shows that some changes in the data assimilation also created discontinuities in the random

error, whose correction is not so simple. The potential implications in other ERA5 snow-related parameters such as surface albedo or hydrological variables have not been evaluated in this study but could be significant in snow-covered regions as well.

Satellite products generally provide more accurate and stable estimates but their temporal coverage is limited to that of the satellite instrument. Different satellite instruments can be combined to extend the temporal coverage of the products, which alters the stability of the product during the transition period. The probability of adding artificial trends increases even more in products that assimilate a non-uniform number of satellite data such as IMS or NOAA CDR, similar to the reanalysis assimilation system. In any case, the temporal coverage of most satellite products is limited to the start of the satellite era. The NOAA CDR was able to extend its coverage up to 1969 by combining observations from different sensors and products with manual processing. This makes it the longest satellite CDR available, and the one typically used in climate studies. However, the present study corroborates the existence of a positive artificial trend around [+5, +10 days/decade] in SON (mostly over Russia) and reveals the presence of a similar trend in DJF (over Europe and Eastern USA). Both trends are most likely related to an improved detection of the snow onset due to the increasing number of satellite data ingested. This artificial trend explains the SCE recovery observed in SON and DJF, which opposes the trends observed with other satellite products and station measurements in these seasons. Therefore, NOAA CDR estimations in these seasons should be corrected to obtain reliable results (e.g., (Hori et al., 2017)). Moreover, using multi-datasets instead of a single product to calculate snow cover trends should be preferred, as also suggested by Mudryk et al. (2020). Note that despite multi-datasets are much more robust, characterizing the stability of the individual products is still critical to obtain stable ensembles, particularly when different products share the same instabilities (e.g., ERA5 and ERA5-Land).

## 5  Conclusions

This study evaluates the temporal stability of ERA5 (1950-2020), ERA5-Land (1950-2020) and NOAA CDR (1968-2020) for analyzing snow trends. Despite being some of the longest satellite and reanalysis datasets available and being extensively used for climate application, the study reveals the existence of different artificial trends/discontinuities in the three products that compromise their temporal stability. In the reanalysis, data assimilation creates a trade-off between accuracy and stability. ERA5 presents the worst temporal stability overall due to three negative step-wise discontinuities caused by the assimilation of new observations, but it shows the best accuracy after 2004 when the amount of data assimilated is the largest. By contrast, ERA5-Land does not assimilate data showing better stability but worse accuracy.

NOAA CDR presents a positive artificial trend in SON and DJF. These results provide another line of evidence supporting the problematic fall trends in NOAA CDR, and reveal that a similar trend appears in Europe of eastern North America during winter. Despite the numerous studies highlighting the inconsistency of NOAA CDR fall trends with in-situ measurements and with other datasets, some studies keep claiming a positive snow cover trend in fall based solely on NOAA CDR data (Cohen et al., 2021). Using NOAA CDR without correction in SON and DJF should be avoided. NOAA CDR could still be valid after correction, or in other regions and seasons (e.g., MAM) not affected by artificial trends.

We also analyze the NH snow trends (1950-2020) based on in situ measurements. The analysis shows a global decrease in SCD driven mostly by an earlier melt in spring, which is directly linked to the snow-albedo feedback. However, a decrease due to a later snow onset in fall is also observed during the last years. In warmer regions such as Europe, SCD decrease is aggravated by a decreasing snow depth, which could be related to the decreasing amount of precipitation as snowfall. In drier regions such as Russia, SCD also decreases (except in Ural region and Sea of Okhotsk) despite the increase in snow depth observed over Russia due to warmer and moister weather.

## Appendix A: Additional figures and tables

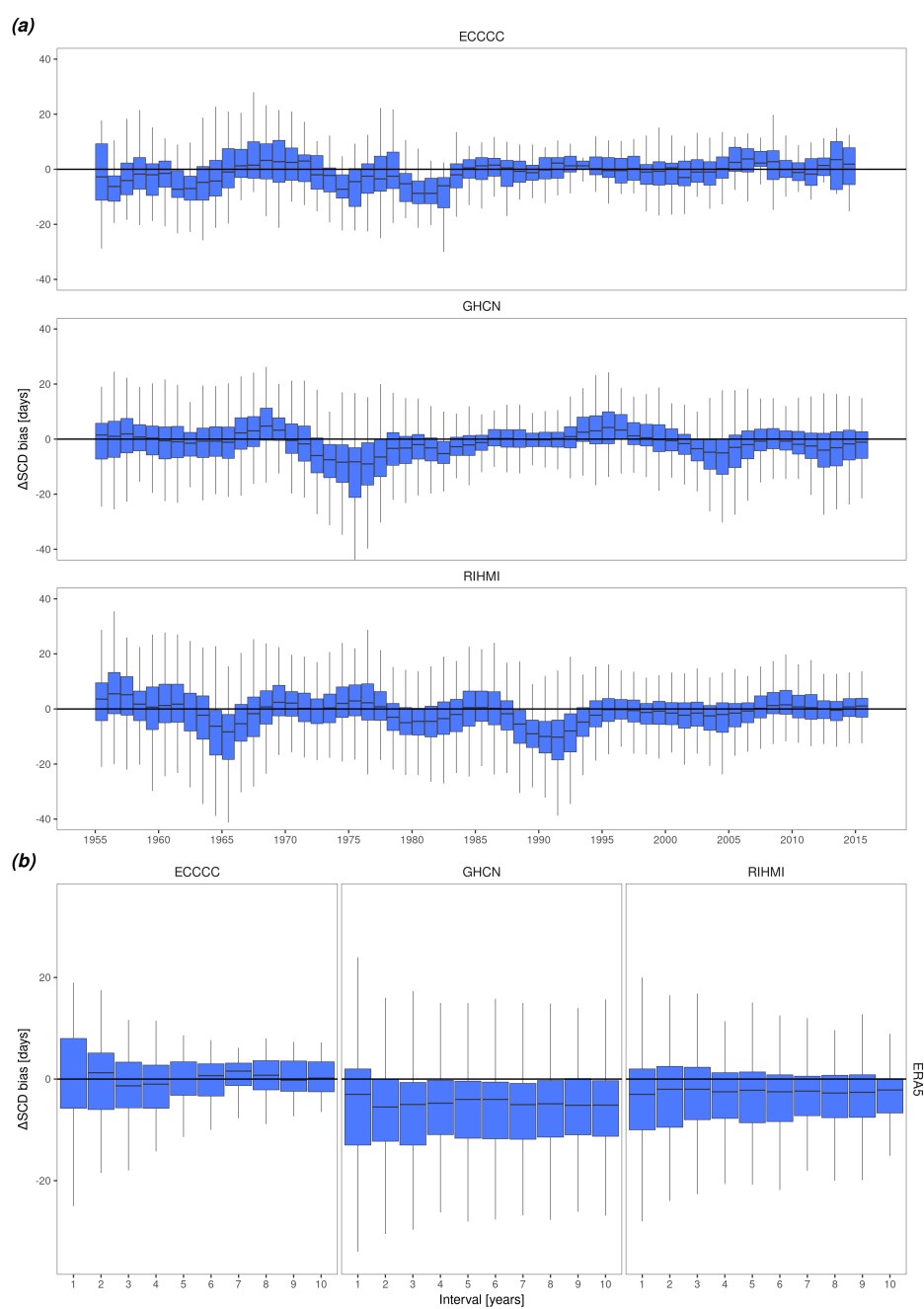

**Figure A1.** Sensitivity analysis to determine **(a)** the exact year of step discontinuities, and **(b)** the interval used to estimate the magnitude of the discontinuity. **(a)** Change in ERA5 $\Delta SCD\ bias$ (before – after) when the step year varies from 1955 to 2015. **(b)** Change in ERA5 $\Delta SCD\ bias$ (before – after) during the 2004 discontinuity when the number of years used for its calculation (interval) is varied from 1 to 10.

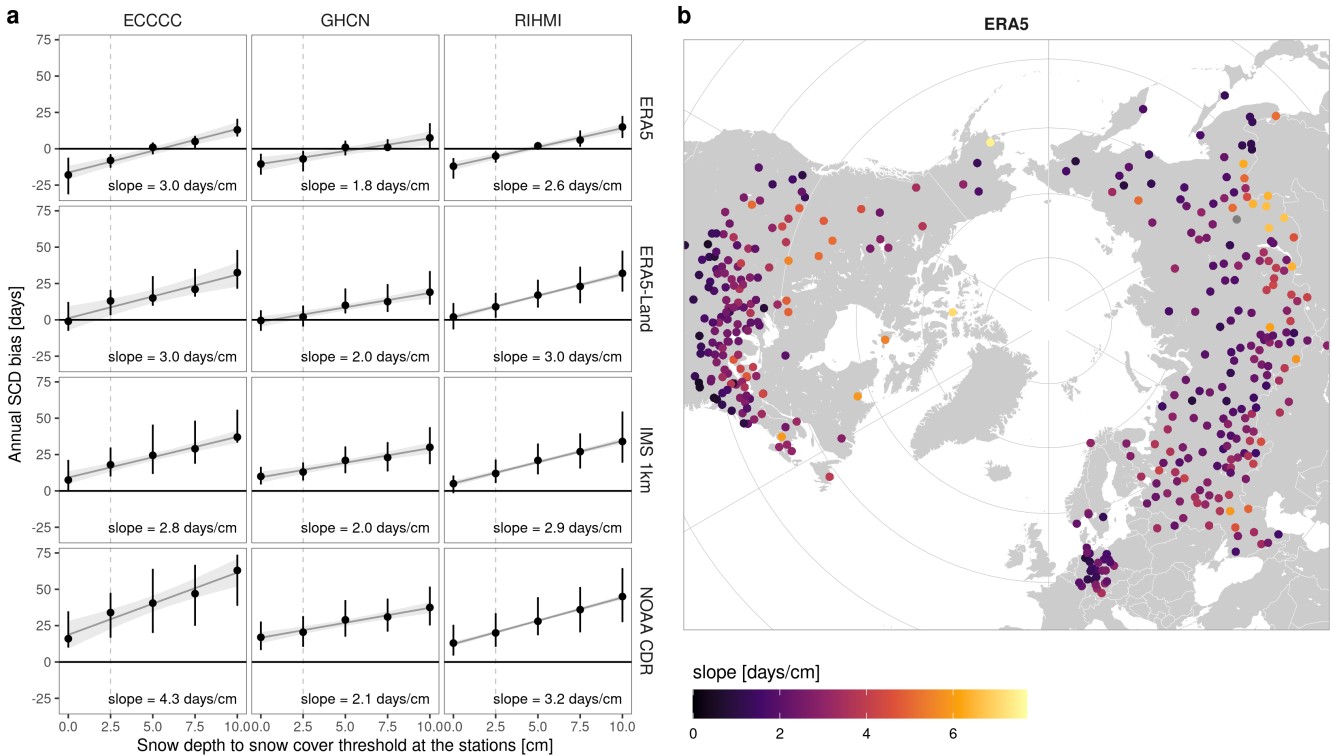

**Figure A2.** Sensitivity of the snow cover duration (SCD) bias on the snow depth to snow cover threshold used at the stations. (**a**) Variation of the SCD bias (median ± interquartile range) per product and network when changing the threshold from 0 to 10 cm by intervals of 2.5 cm. (**b**) Spatial analysis of the rate of change [days/cm] for ERA5. Both figures are derived with data from 2015.

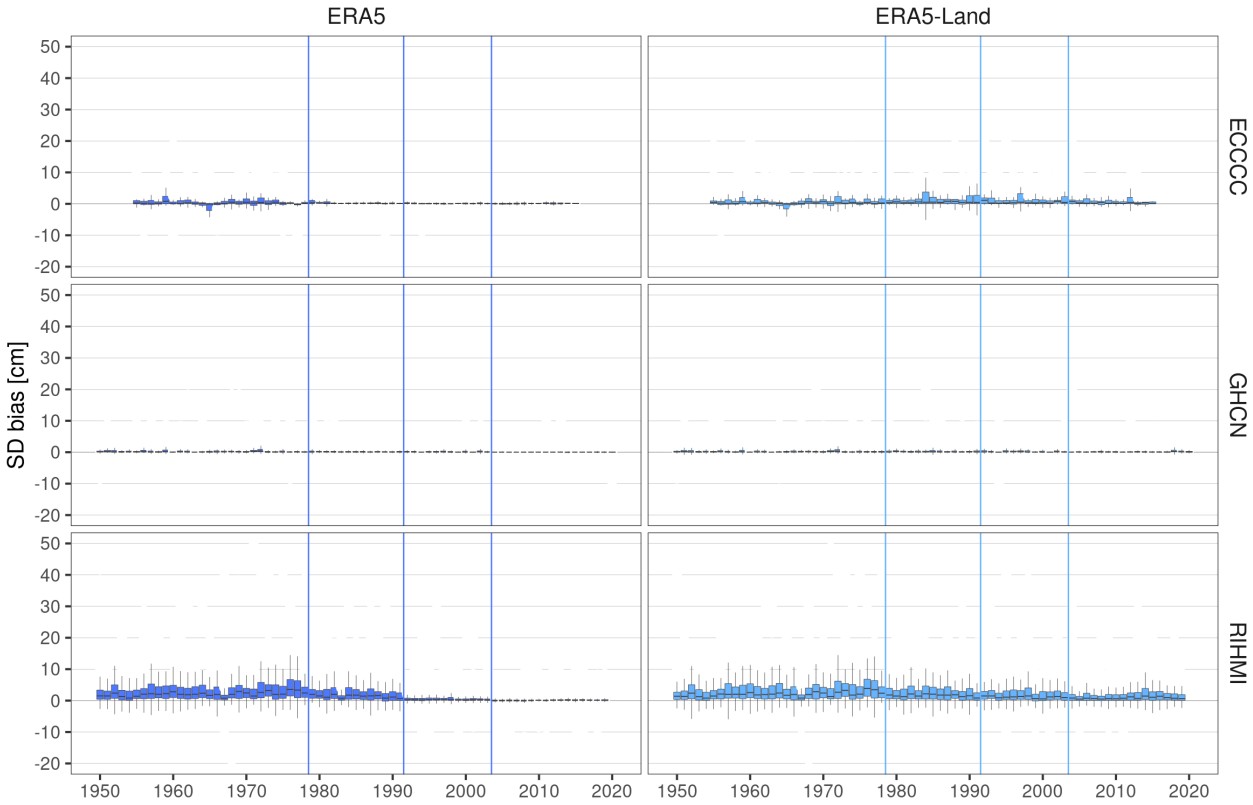

**Figure A3.** Temporal stability of the bias (product - station) in snow depth (SD) per product and network during SON. Vertical lines show the years when the potential discontinuities in each product occur.

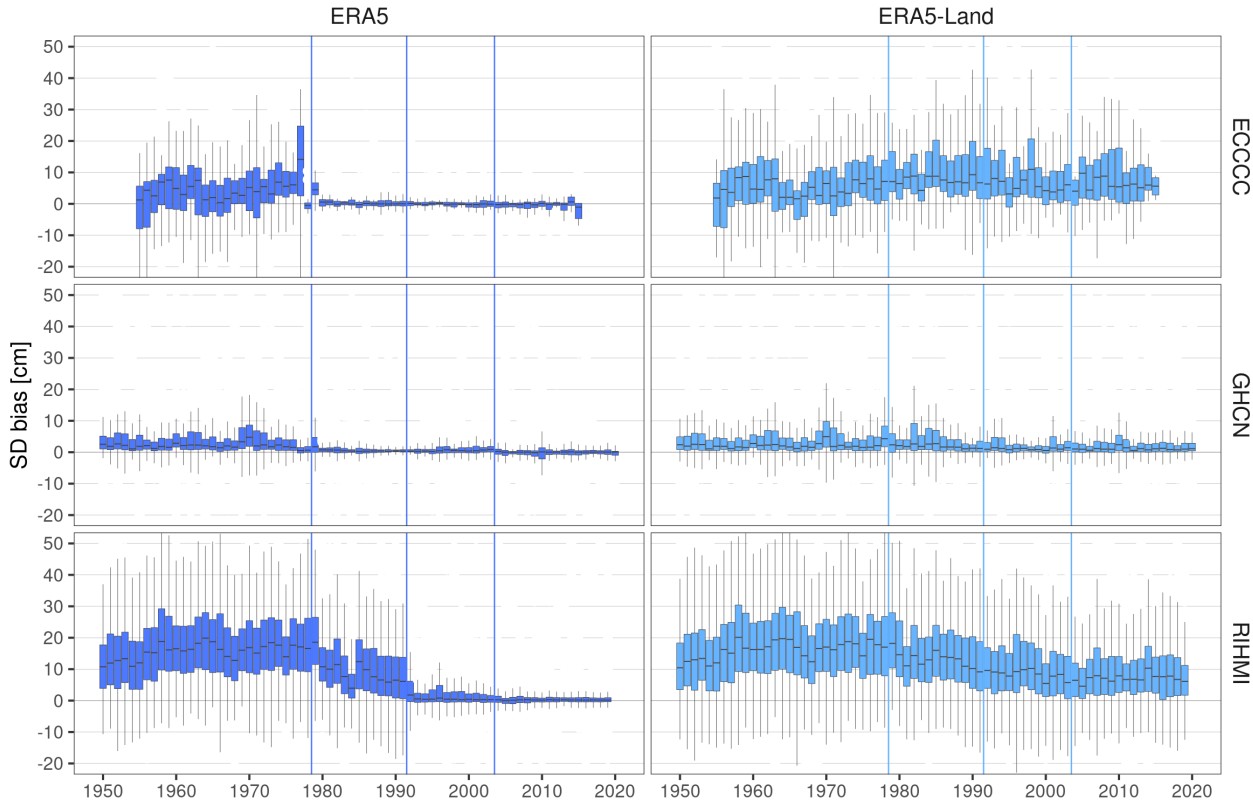

**Figure A4.** Temporal stability of the bias (product - station) in snow depth (SD) per product and network during DJF. Vertical lines show the years when the potential discontinuities in each product occur.

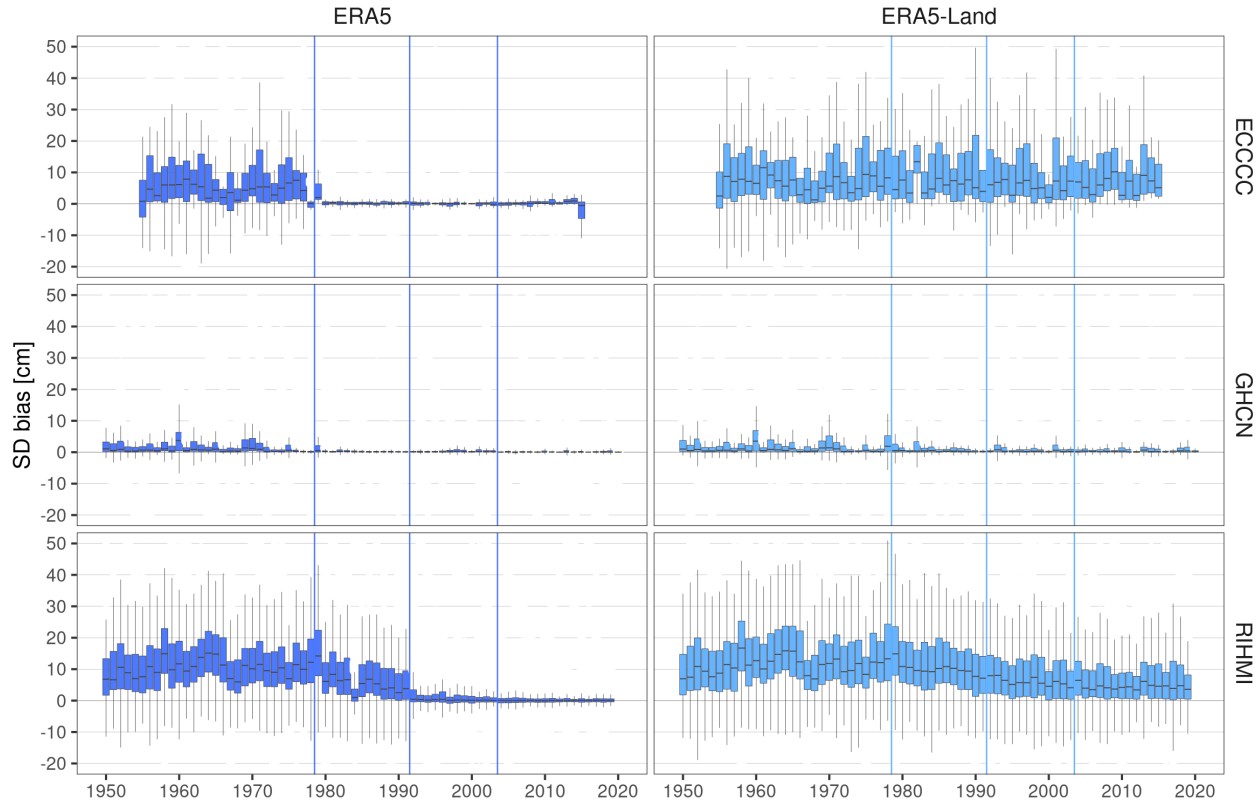

**Figure A5.** Temporal stability of the bias (product - station) in snow depth (SD) per product and network during MAM. Vertical lines show the years when the potential discontinuities in each product occur.

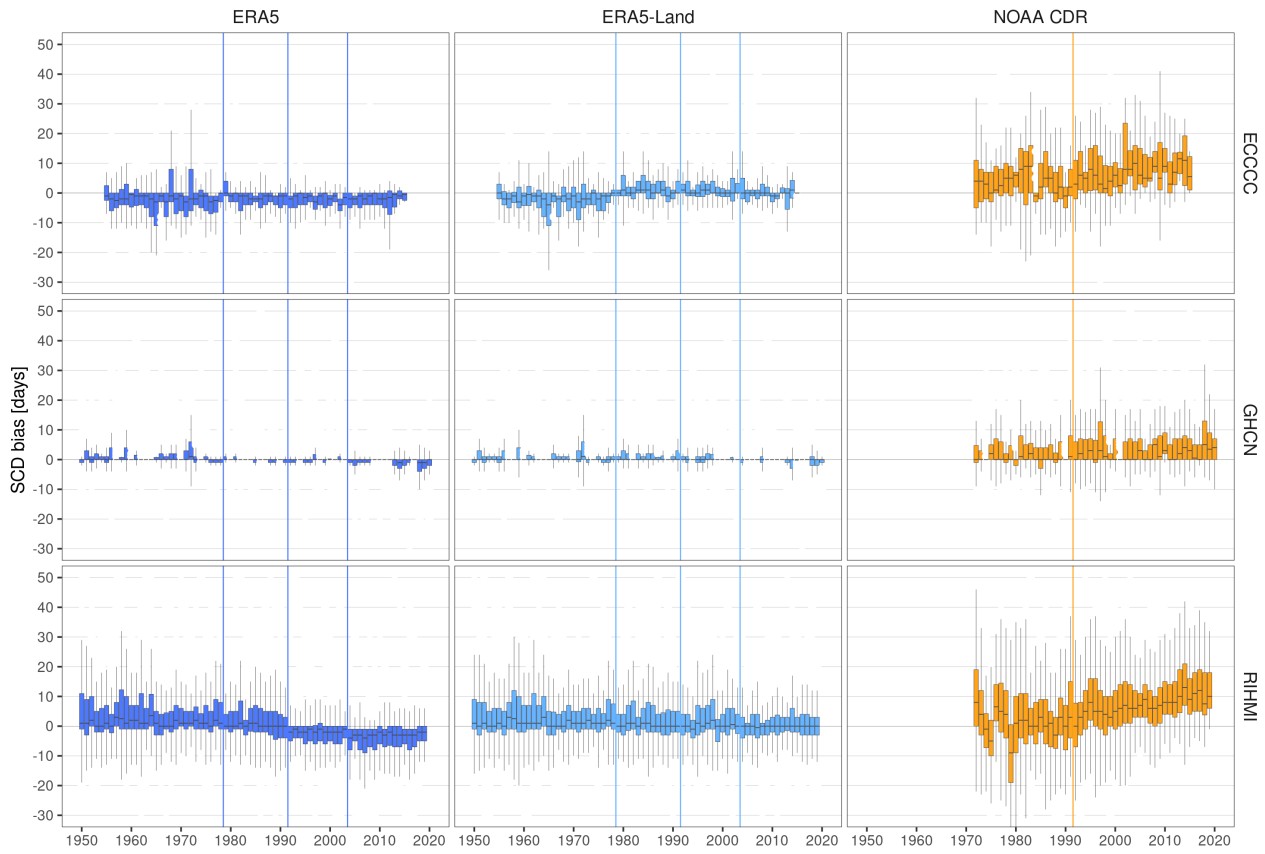

**Figure A6.** Temporal stability of the bias (product - station) in snow cover duration (SCD) per product and network during SON. Vertical lines show the years when the potential discontinuities/trends in each product occur/start.

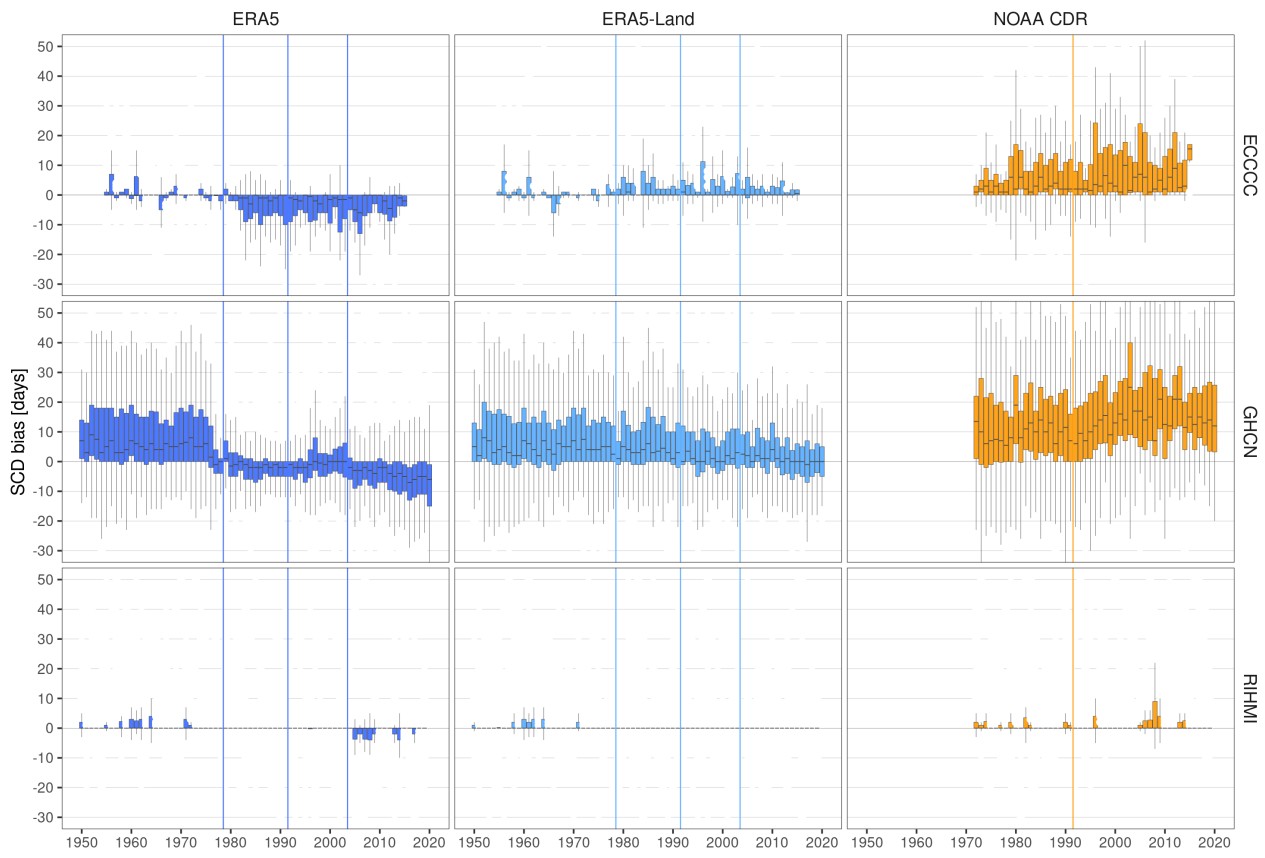

**Figure A7.** Temporal stability of the bias (product - station) in snow cover duration (SCD) per product and network during DJF. Vertical lines show the years when the potential discontinuities/trends in each product occur/start.

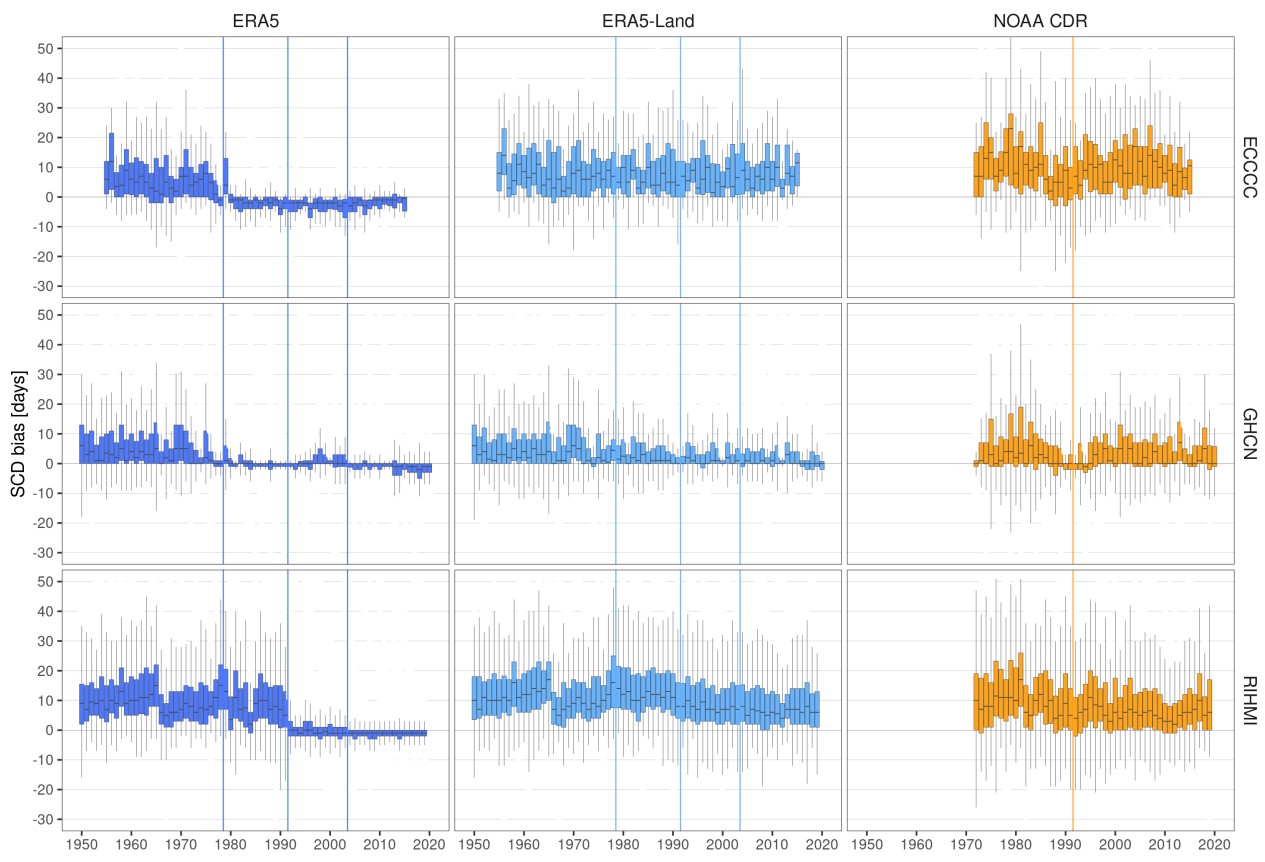

**Figure A8.** Temporal stability of the bias (product - station) in snow cover duration (SCD) per product and network during MAM. Vertical lines show the years when the potential discontinuities/trends in each product occur/start.

**Table A1.** Change in the ERA5 bias (median with its 95 % CI) during 1977-80, 1991-92 and 2004-05 discontinuities. The four years before and after the discontinuity are compared ($\Delta bias = bias_{after}$ - $bias_{before}$). Valid[%] depicts the percentage of stations that meet the GCOS stability requirements

| | | | SD | | | SCD | |
|---|---|---|---|---|---|---|---|
| | | | $\Delta bias[cm]$ | $\Delta bias[\%]$ | Valid [%] | $\Delta bias[days]$ | $\Delta bias[\%]$ |
| 1977-80 | ECCC | Annual | -3.5 [-5.1, -2.3] | -49.8 [-67.1, -17.9] | 9.1 | -8.2 [-13.5, -3.8] | -4.8 [-11.7, -1.4] |
| | | SON | 0 [0, 0] | 0.6 [0, 1.4] | 75.8 | 1 [0.2, 1.8] | 3.9 [1.5, 6.3] |
| | | DJF | -8.5 [-9.8, -7.3] | -126.4 [-228.1, -65.1] | 0.0 | -0.8 [-1.2, 0] | -1.8 [-6.2, 0] |
| | | MAM | -5.8 [-7.9, -4.6] | -104.3 [-188.9, -71.6] | 15.2 | -7.5 [-7.8, -5] | -22.6 [-36.4, -12.9] |
| | GHCN | Annual | -0.3 [-0.5, -0.2] | -24 [-35.1, -16.3] | 74.1 | -7.5 [-10.2, -6] | -22.5 [-27, -17.3] |
| | | SON | 0 [0, 0] | 3.3 [1.8, 6.9] | 96.2 | 0.2 [0, 0.2] | 1.2 [0, 2.8] |
| | | DJF | -0.9 [-1.2, -0.7] | -145.8 [-274.8, -92.4] | 41.1 | -6.4 [-7.4, -5.2] | -152.3 [-216.7, -108] |
| | | MAM | -0.1 [-0.2, -0.1] | -13.7 [-25.8, -8.6] | 73.4 | -1 [-1.5, -0.8] | -16.7 [-26.3, -11.1] |
| | RIHMI | Annual | -2.3 [-2.8, -1.8] | -19.4 [-25.9, -14.9] | 19.7 | -2.5 [-3.5, -0.8] | -1.5 [-2.1, -0.4] |
| | | SON | -1 [-1.1, -0.9] | -9.6 [-13, -7.5] | 39.9 | -0.3 [-0.8, -0.1] | -1.1 [-2.5, -0.2] |
| | | DJF | -5.8 [-6.2, -5.3] | -64.3 [-82.7, -50.8] | 10.6 | 0 [0, 0] | 0 [0, 0] |
| | | MAM | -1.8 [-2, -1.4] | -19.7 [-25.3, -13.3] | 20.7 | -0.5 [-1.2, -0.2] | -1.8 [-2.9, -0.3] |
| 1991-92 | ECCC | Annual | -0.1 [-0.2, 0] | -0.9 [-3, 1.1] | 84.8 | 2.2 [-3, 3.5] | 1.6 [-2, 2.7] |
| | | SON | 0 [0, 0] | 0.2 [0, 1.1] | 100.0 | 0.5 [0.2, 0.7] | 1.4 [0.5, 2.1] |
| | | DJF | 0 [-0.3, 0.1] | -0.7 [-4.9, 1] | 69.7 | 0 [0, 0.5] | 0 [0, 1.5] |
| | | MAM | -0.1 [-0.2, -0.1] | -2.5 [-5.9, -0.5] | 75.8 | 0 [-0.5, 0.5] | 0 [-0.9, 1.3] |
| | GHCN | Annual | 0 [0, 0] | -0.1 [-3, 4.4] | 88.0 | 0.2 [-0.8, 1] | 0.6 [-1.8, 2.7] |
| | | SON | 0 [0, 0] | 0.3 [-0.6, 1.6] | 95.6 | 0.1 [0, 0.2] | 1.5 [0, 3.2] |
| | | DJF | 0 [-0.1, 0] | -2.9 [-9.9, 4.6] | 72.2 | 0 [0, 0.5] | 0.4 [0, 5.6] |
| | | MAM | 0 [0, 0] | 1.4 [-0.3, 4.4] | 83.5 | 0 [0, 0] | 0 [-1.3, 0] |
| | RIHMI | Annual | -2.2 [-2.8, -1.6] | -18 [-26.1, -12.9] | 24.9 | -10.2 [-12, -8.5] | -6.3 [-7.7, -5.3] |
| | | SON | -0.8 [-0.9, -0.6] | -7.9 [-10, -5.9] | 48.1 | -1.8 [-2.2, -1.2] | -5.4 [-7, -4.2] |
| | | DJF | -4.8 [-5.7, -4] | -52 [-67.8, -38.4] | 12.2 | 0 [0, 0] | 0 [0, 0] |
| | | MAM | -3 [-3.5, -2.7] | -34 [-45.8, -26.9] | 17.5 | -6.8 [-7.2, -6.2] | -20.4 [-23.6, -17.9] |
| 2004-05 | ECCC | Annual | 0 [-0.2, 0.1] | -0.1 [-2.6, 1.2] | 93.9 | -1 [-5.2, 2] | -0.8 [-3.3, 1.7] |
| | | SON | -0.1 [-0.1, 0] | -0.8 [-1.3, 0.2] | 93.9 | 0.1 [0, 0.5] | 0.5 [0, 1.9] |
| | | DJF | -0.1 [-0.5, 0] | -2.1 [-4.3, 0] | 69.7 | -1.2 [-2.2, 0] | -3.3 [-6.2, 0] |
| | | MAM | 0.1 [-0.1, 0.2] | 1.2 [-0.8, 2] | 87.9 | 0 [-0.2, 0.7] | 0 [-1.2, 1.6] |
| | GHCN | Annual | -0.2 [-0.3, -0.1] | -16.8 [-25.1, -10.9] | 79.2 | -4.8 [-6.8, -2.8] | -15.6 [-20.6, -9.1] |
| | | SON | -0.1 [-0.1, -0.1] | -30.1 [-51.9, -17.9] | 94.8 | -0.5 [-0.8, -0.5] | -15.9 [-26.3, -10] |
| | | DJF | -0.5 [-0.7, -0.4] | -119.8 [-178.9, -85.4] | 50.0 | -2.8 [-3.8, -2] | -48.2 [-75, -33.3] |
| | | MAM | -0.1 [-0.2, -0.1] | -47.8 [-68.4, -29.1] | 76.6 | -0.8 [-1.2, -0.5] | -23.7 [-37.2, -14] |
| | RIHMI | Annual | -0.1 [-0.1, 0] | -0.6 [-1.3, -0.2] | 74.1 | -2.5 [-3.5, -1.2] | -1.5 [-2.2, -0.6] |
| | | SON | -0.2 [-0.2, -0.2] | -3.2 [-4.3, -2.5] | 78.8 | -1 [-1.5, -0.5] | -3.3 [-4.4, -2.1] |
| | | DJF | 0 [-0.1, 0] | -0.5 [-1, -0.1] | 58.2 | 0 [0, 0] | 0 [0, 0] |
| | | MAM | -0.1 [-0.1, 0] | -0.6 [-1.2, -0.3] | 67.7 | -0.5 [-0.8, -0.2] | -1.2 [-1.8, -0.7] |

**Table A2.** Decadal trend (median with its 95 % CI) of the seasonal bias in snow cover duration (SCD) of NOAA CDR (1992-2015) per region. $N$ shows the number of stations showing significant trends (p < 0.05, Mann-Kendall). Only statistically significant trends are included in the median calculation.

| | SON | | DJF | |
|---|---|---|---|---|
| | Trend [days/decade] | N | Trend [days/decade] | N |
| ECCC E | - | 2 (10.5%) | - | 2 (10.5%) |
| ECCC N | - | - | - | 2 (100%) |
| GHCN USA-W | - | 3 (15%) | - | 3 (15%) |
| GHCN USA-E | 3.2 [0.5, 7.4] | 7 (6.9%) | 7.1 [6, 10.1] | 20 (19.6%) |
| GHCN NO | - | - | - | 2 (50%) |
| GHCN EU | - | 2 (7.4%) | 8.3 [7.5, 10.2] | 6 (22.2%) |
| RIHMI EU | 8.8 [5.5, 11] | 14 (34.1%) | 6.3 [4.7, 8.8] | 19 (46.3%) |
| RIHMI Ural | 6.4 [5.8, 7.1] | 17 (31.5%) | 3.3 [2.1, 7.5] | 6 (11.1%) |
| RIHMI Siberia | 5.6 [4.6, 7.3] | 12 (29.3%) | - | - |
| RIHMI S | 8.6 [4.4, 15.7] | 6 (18.2%) | - | - |
| RIHMI E | 5.9 [5, 10.4] | 6 (30%) | - | - |

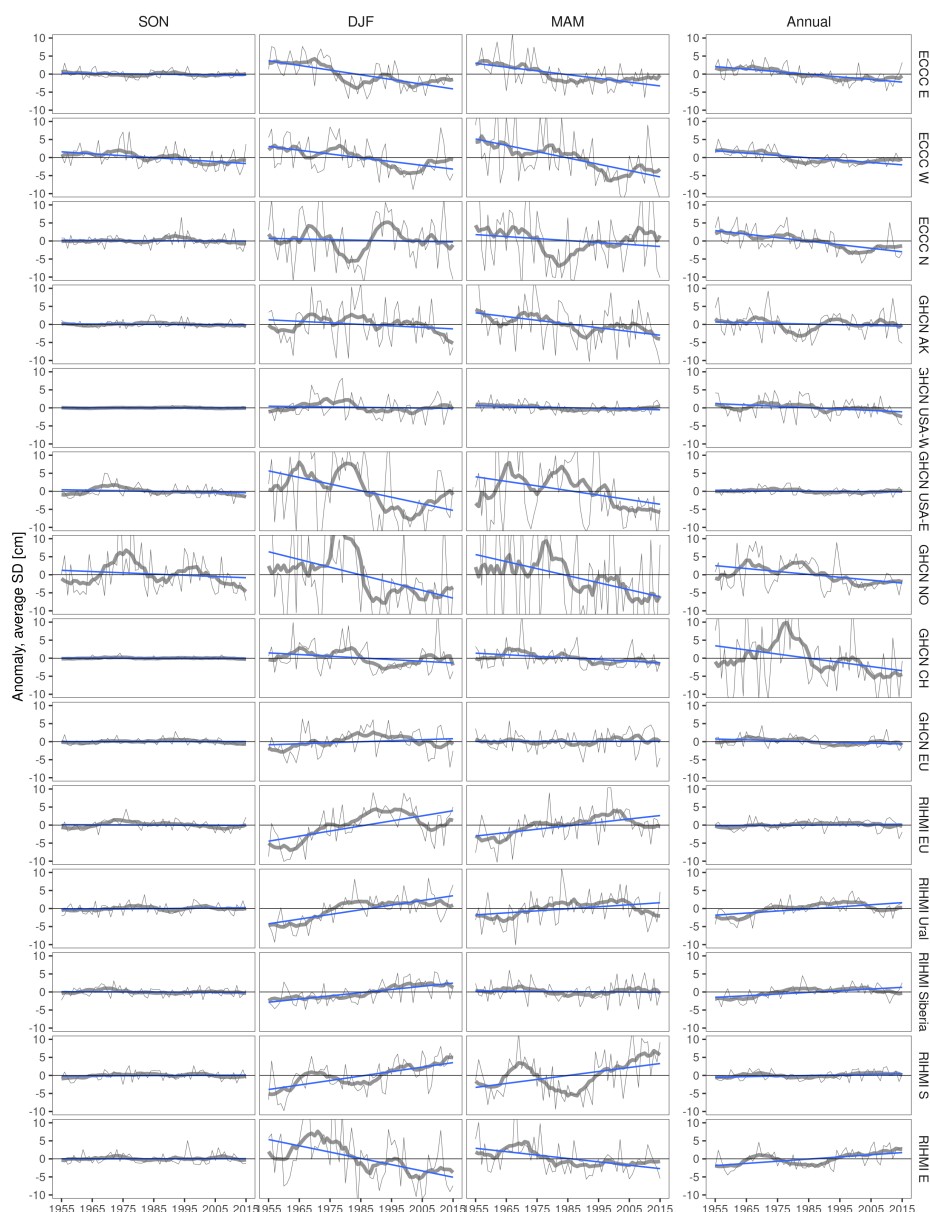

**Figure A9.** Annual and seasonal anomalies in snow depth (SD) per spatial region compared to the 1955-2015 reference period. Grey think lines show the 10-year running mean.

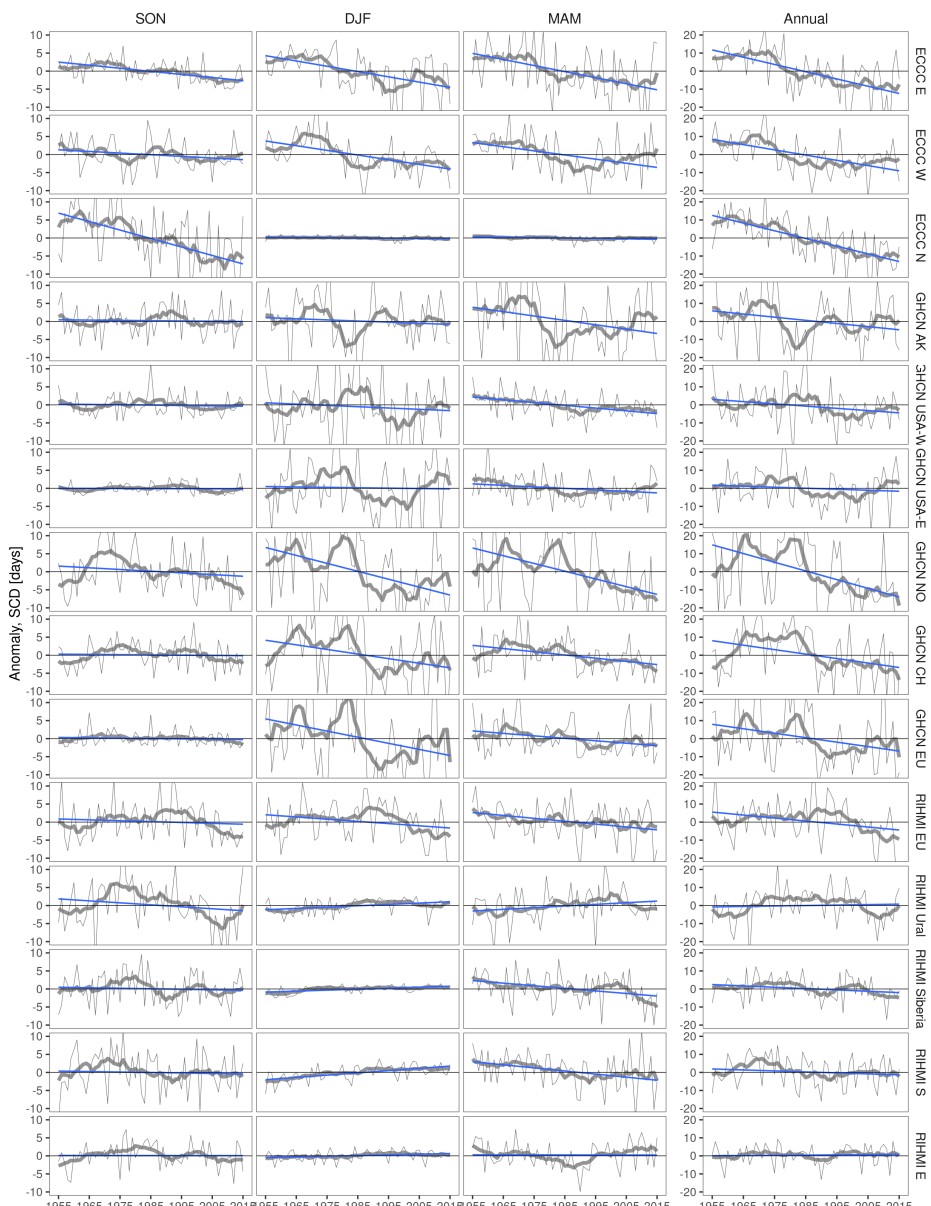

**Figure A10.** Annual and seasonal anomalies in snow cover duration (SCD) per spatial region compared to the 1955-2015 reference period. Grey think lines show the 10-year running mean.

*Author contributions.* RU designed the experiment, performed the analysis and wrote the original manuscript. NG supervised the study and reviewed the document. All authors have read and agreed to the published version of the manuscript.

*Competing interests.* The authors declare that they have no conflict of interest.

*Acknowledgements.* The support provided by DG DEFIS, i.e. the European Commission Directorate General for Internal Market, Industry, Entrepreneurship and SMEs, and Copernicus Programme is gratefully acknowledged. ERA5 and ERA5-Land were retrieved from the Copernicus Climate Data Store (CDS, https://cds.climate.copernicus.eu/#!/home). The NOAA CDR snow cover product was retrieved from the NH SCE version 4 available at NSIDC (https://nsidc.org/data/NSIDC-0046/versions/4). NOAA's IMS 1 km snow cover product used to evaluate the spatial representativeness was also retrieved from NSIDC (https://nsidc.org/data/G02156/versions/1). We also acknowledge the networks of ground stations, GHCN and RIHMI-WDC, for maintaining and providing the daily in-situ measurements of snow depth used in the study.

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
