# Peer review of "Temporal stability of long-term satellite and reanalysis products to monitor snow cover trends"

_The Cryosphere, 2021_

## Author Comment (AC1)

**Answers to reviewers: TC-2021-281**

**Temporal stability of long-term satellite and reanalysis products to monitor snow cover trends**

Ruben Urraca and Nadine Gobron

**REFEREE #1 – Chris Derksen**

This study uses a reference dataset of point snow depth measurements to assess the performance and stability of snow extent and snow cover duration from reanalysis and satellite-derived products. This is important to quantify because changes to the quality and quantity of satellite data and the data sources assimilated into reanalysis can introduce spurious trends and temporal discontinuities into multi-decadal time series. The analysis is focused on ERA5 and the NOAA snow chart climate data record (NOAA-CDR), which are two widely used datasets that provide snow information back to the 1960s. Overall, I found the analysis to be comprehensive in scope, sound in the overall approach, and clearly explained.

I have a number of both major and minor comments, mostly in an effort to further clarify the methods and tighten the messaging. This was a really enjoyable paper to review, thanks to the authors for their efforts.

**MAJOR COMMENTS**

Lines 61-68: Some additional context/examples could be provided in this paragraph.

First: "The transition between different sensors (e.g., JAXA GHRM5) or increasing the number of satellite sources used (e.g., IMS, NOAA CDR)…" It may not be clear to some readers that the IMS product is actually manually derived by analysts from multiple sources of satellite imagery (as opposed to an objective retrieval like the JAXA product). This is noted later on line 88, but this could be mentioned in this introductory paragraph.

Second: The ESA GlobSnow and Snow CCI products are derived from the passive microwave satellite record, which is composed of SMMR + SSM/I + SSMIS data, which is another example of how discontinuities can be introduced through changing instruments during the satellite era. (Incidentally, we have found there are differences in the validation statistics for Snow CCI SWE performance related to the different passive microwave sensors. This work is under review, but it would be interesting to also include the Snow CCI dataset in the analysis you present in this work.)

**Answer**: Thanks for the comments. We have rephrased the paragraph including the reviewer's suggestions:

*"The temporal coverage of satellite products is limited by the satellite/sensor used, so different satellite instruments are combined to produce Climate Data Records (CDRs). For instance, JAXA GHRM5 combines optical data from NOAA's AVHRR and MODIS sensors, whereas both ESA GlobSnow and ESA snow CCI SWE combine passive microwave data from SMMR, SSM/I and SSMIS sensors. The transition periods between different sensors are the main source of instability in these products, but stability issues can also arise due to sensor degradation and orbital drifts (e.g., AVHRR data). The increasing number of satellite sources can also alter the stability of products derived manually by analysts from multiple sources of satellite imagery (e.g., IMS and NOAA CDR)."*

**Figure 1:** It's unfortunate no data from Canada were used in this study (particularly in the context of the trend analysis in Figure 9, which gives the impress of negative trends in the Eurasian sector and no trends over Canada, which is not the case). There is an updated snow depth dataset for Canada described here: Brown, R., C. Smith, C. Derksen, and L. Mudryk. 2021. Canadian in situ snow cover trends 1955-2017 including an assessment of the impact of automation. Atmosphere-Ocean. DOI: 10.1080/07055900.2021.1911781. For future reference, the Canadian Historical Daily Snow Depth Database should soon be available here (or contact the authors of the above paper): https://catalogue.ec.gc.ca/geonetwork/srv/eng/catalog.search#/metadata/63dca4bb-a29a-43b0-828b-7eccb03de456

**Answer**: Many thanks for providing us the link to the Canadian snow cover dataset. We agree that Canada was the main spatial gap in our study, particularly for trend analysis. We have processed all the stations available in the Canadian Historical Daily Snow Depth Database. Out of them, 57 passed our selection criteria for trend/stability analysis, and 34 were classified as spatially representative for the stability assessment.

As mentioned by Brown et al 2021, the number of Canadian stations significantly decreases before 1955 and after 2010.  If we kept our original study period for the trend analysis (1950-2020), the number of Canadian stations available drops below 10. Therefore, to cover most Canadian regions, we have reduced the study period for the trend analysis from 1950-2020 to 1955-2015.

**Section 2.2:** How did you ensure that the snow depth observations retained for analysis were not assimilated into ERA5? This issue must be addressed specifically in the text to ensure independence between the reanalysis and validation datasets.

**Answer**: C3S/ECMWF currently does not provide the list of snow stations assimilated by ERA5. This information could be included in future updates of ERA5. ECMWF specifies that ERA5 assimilates SYNOP stations, and some RHIMI, GHCN and ECCC stations may report to the SYNOP network as well. Thus, some stations used for the validation could be currently assimilated by ERA5.

We acknowledged this issue in our original submission, but we have extended the discussion on the potential limitations of our study in the revised version.

*"ERA5 assimilates 3507 snow depth observation from SYNOP, but the exact list of stations assimilated is not yet available (ECMWF, personal communication). It is likely that some RHIMI, GHCN, and ECCC stations were assimilated by ERA5, since some stations of these networks also report to SYNOP. This could compromise the independence of our validation set, particularly for the spatial accuracy analysis of ERA5 between 2005-2015 (Section 3.3). Particularly, the magnitude of the bias and RMSE will be artificially reduced at those stations assimilated by ERA5. However, the impact of this issue in stability analysis, which is the main goal of the study, should be smaller. The step changes and trends observed in the ERA5 bias do exist and are due to changes in the ERA5 model, independently of whether some stations are assimilated or not by ERA5. It is true that the assimilation of the stations could explain some of the step changes observed in ERA5 (e.g., RHIMI 1991). In this case, discontinuities will be higher at those stations assimilated by ERA5, but they will also appear at stations not being assimilated."*

**Section 2.4: Very interesting comparison with the analysis of Hori et al (2017). I'm not fully clear on how the SCF surrounding the station was determined: "In this study, we used the SCF in the surroundings of the station measured at RIHMI stations to analyze the correlation between SD at the station and the surrounding SCF (Fig. 2)." Was IMS data used to determine SCF? What distance was used around the station (IMS pixels contained by coarser ERA5/ERA5-Land pixels as described in Section 2.3?)?**

**Answer**: The snow cover fraction around the station is visually assessed at RIHMI stations. The exact definition of these measurements is given by Bulygina et al 2011:

*"The snow cover extent over the near station territory and the snow cover characteristics are visually determined at morning observations. The amount of snow covering the visible area around a meteorological station is estimated on a scale of one to ten (10–100%; or zero in the absence of snow)."*

We have extended the description of the SCF around the station in the methods section, to clarify that it is a visual measurement made at the stations:

*"The SCF in the surrounding of the stations is visually assessed at RIHMI stations (Bulygina et al 2011). We used these measurements to analyze the correlation between SD at the station and the surrounding SCF (Fig.2)."*

**Section 2.4.1: It is noted that "Stability was evaluated by analyzing how the annual bias in both SD and SCD changed temporally." and that stability was analyzed separately for the RIHMI and GHCN networks. But how were step changes statistically determined (the vertical lines in Figures 4 and 5)? Line 198: Why was the interval of four years selected to compare the bias difference before and after a step discontinuity? Was there any testing performed to confirm that this was some sort of ideal number?**

**Answer**: Some vertical lines (years of step changes) are determined by significant changes in the product algorithm: 1979 corresponds to the transition between ERA5 and ERA5 backward extension, 2004 corresponds to the addition of IMS snow data to ERA5 model. The exact year of other discontinuities (e.g., ERA5 1991) was determined with a window function that calculated how the magnitude of the step changes when varying the step year from 1950 to 2020 by intervals of 1 (Fig A1). The relative maximum or minimum was selected as the discontinuity year.

[Figure]

**Fig A1** Sensitivity analysis to determine the exact year of step discontinuities **(a)** and the interval used to estimate the magnitude of the discontinuity **(b). (a)** Change in ERA5 SCD Δbias (before – after) when the step year varies from 1955 to 2015. **(b)** Change in ERA5 SCD Δbias (before – after) during the 2004 discontinuity when the number of years used for its calculation (interval) is changed from 1 to 10.

The interval of 4 years used to compare the bias after and before the step was a compromise between two effects.

- The interval should be long enough to remove the effects of inter-annual snow cover variability on Δbias.

- The interval should be as short as possible to remove the effect of underlying trends in the bias on Δbias.

We evaluated the magnitude of both effects with a sensitivity analysis, measuring how the magnitude of Δbias varied with an increasing interval, from 1 to 10 years by 1-year intervals. The results show that Δbias variability stabilizes after 4-5 years. Therefore, we used an interval of 4 years

We have added this figure as supplementary material, and we have included the following explanation in the methods section:

*"The interval of four years was chosen based on a sensitivity analysis (Fig A1). This interval needs to be long enough to remove the effects on inter-annual variations of the snow cover, but too long intervals may be affected by the underlying trends in the bias. Therefore, the shortest interval after $\Delta$bias has stabilized was chosen."*

**Section 2.5: What is the justification for including the stations which failed the spatial representative test in the trend analysis?**

**Answer**: The main goal of the spatial representativeness test is to reduce the uncertainty of the point-to-pixel comparison, discarding stations in which point measurements are not representative of the larger surrounding region covered by the reanalysis pixel. However, stations discarded for the point-to-pixel validation are still valid for the trend analysis.

For instance, most of the stations classified as low representative are in coastal regions (Fig 3). However, snow cover trends at these coastal stations are still meaningful for the trend analysis. Indeed, some of these coastal regions such as Eastern USA and Eastern Canada are those experiencing a larger snow cover retreat. Besides, trends observed at coastal stations are coherent with those observed at inland locations. Thus, we believe that keeping these stations provides additional valuable information to trend analysis, without interfering with the representatives of this analysis.

**Section 3.1: I appreciate the effort taken to quantify the spatial representativeness of the point measurements. This is a long standing problem in the validation of gridded snow products at variable resolutions, which is usually acknowledged but not addressed analytically. So these results are very interesting…**

**Answer**: Many thanks for the comment.

**Line 291: "This suggests that the H-TESSEL land model used in both ERA5 and ERA5-Land tends to systematically overestimate SD, most likely due to an excessive snowfall, when no data is assimilated (ERA5 before 1979, ERA5 above 1500 m, ERA5-Land)." I find the messaging in this sentence to be confusing. If the overestimation is related to H-TESSEL, this implies that uncertainty**

**in snow parameterizations in the model lead to overestimation of snow depth, but then the problem of excessive snowfall is mentioned. Does this not imply that precipitation bias is the source of the positive SD bias as opposed to the land model?**

**Answer**: We have rephrased this sentence to clarify that the most likely cause of the SD bias is a precipitation bias, as suggested by Orsolini et al 2019:

*"As suggested by Orsolini et al 2019, the most likely cause of the snow depth overestimation in both ERA5 and ERA5-Land could be a precipitation bias, which is only corrected by the assimilation of snow depth observations in ERA5 (after 1979 and below 1500 m)."*

**Section 3.2.2: The bias trend in the NOAA CDR in fall is an important finding, and corroborates previous work which found similar issues with this product in this season. This is important because numerous studies continue to cite a positive trend in October snow extent over Eurasia, despite increasing multiple lines of evidence (this study provides a new line of evidence) which outline inhomogeneity in the NOAA CDR. I found lines 335-340 to be somewhat confusing, and suggest this text be edited for clarity. The study of Mudryk et al (2017) could also be considered, which showed (1) the NOAA CDR trends in October and November are non-physical and not consistent with other datasets, and (2) NOAA CDR trends in spring are stronger than other datasets. (Mudryk, L., P. Kushner, C. Derksen, and C. Thackeray. 2017. Snow cover response to temperature in observational and climate model ensembles. Geophysical Research Letters. 44, doi:10.1002/2016GL071789.)**

**Answer**: Tanks, we have rephrased the paragraph adding the new references:

*"Brown and Derksen (2013) suggested that the opposite effect during the spring season could be expected but was not observed. Theoretically, an improved detection of snow melting could lead to a stronger spring trend, introducing an artificial negative trend in the CDR. In this line, Derksen (2014) reported a tendency of NOAA CDR to map less snow in spring since 2007 than the multi-dataset composed by NOAA CDR, MERRA and ERA-Interim. Mudryk et al. (2017) also found that NOAA CDR has a stronger spring trend than that of other datasets. We analyzed this issue by evaluating the snow cover duration trends in spring. Negative trends in spring bias only appear at some Russian stations (Fig.7a). However, the number of stations showing significant trends in spring is smaller, and the magnitude of these trends is much lower than those in fall and winter. Despite this issue could exist in some specific regions, the impact at global scale is negligible (Fig.A8)."*

**Section 3.3: "Both ERA5 and ERA5-land use a threshold (5 cm) larger than the one applied to the stations (2.5 cm)…" In reading Section 2.4, I was wondering about the impact of these different thresholds on the validation analysis. I understand the decision to apply 2.5 cm to the snow depth measurements because this is supported by Figure 2 and is consistent with Hori et al (2017). But calculating bias with slightly different thresholds to convert SD to SCD seems problematic. Can you report on any sensitivity analysis which determines how the bias calculations are related to the choice of threshold as applied to the snow depth measurements?**

**Answer**: Thanks for the suggestion. We have included the sensitivity analysis requested by the reviewer in the results section. We evaluate the changes in snow cover duration bias when changing the station threshold from 0 to 10 by intervals of 2.5 cm.

[Figure]

**Figure A2.** Sensitivity of the snow cover duration (SCD) bias on the snow depth to snow cover threshold used at the stations. (a) Variation of the SCD bias (median ± interquartile range) per product and network when changing the threshold from 0 to 10 cm, by intervals of 2.5cm. (b) Spatial analysis of the rate of change [days/cm] for ERA5. Both figures are derived with data from 2015.

We have discussed the results of the sensitivity analysis in the Results section:

"*In SCD, ERA5 presents a constant underestimation (IQR) of around [-9.4, -5.5 days] while ERA5-Land keeps overestimating [2.4, 11.2 days]. As above mentioned, the SCD bias strongly depends on the threshold used to convert SD to SC. Both ERA5 and ERA5-land use a threshold (5 cm) larger than the one applied to the stations (2.5 cm). This could explain why ERA5 has a negative SCD bias despite having an unbiased snow depth. Indeed, when the ERA5 threshold is applied to the stations (Fig. A2), ERA5 SCD bias is close to zero in the three networks. We could be tempted to use the same threshold in stations and product. However, the thresholds applied by products need to be validated as well, and we can only do it deriving independent thresholds for the station measurements. In this study, we have used visual snow cover measurements made at RHIMI stations, but other data sources such as high-resolution satellite imagery could also be useful.*

*We investigated further this issue with a sensitivity analysis that evaluates how the SCD bias changes with different values of snow depth to snow cover threshold during 2015 (Fig.A2). The magnitude of SCD bias is similar between networks, indicating a good consistency between their measuring methods. However, the magnitude of SCD bias strongly varies between products. When a threshold of 2.5 cm is used, the mean SCD bias varies as follows: 24.8 days (NOAA CDR), 14.3 days (IMS), 8.0 days (ERA5-Land) and -6.7 days (ERA5). These differences are the result of the different thresholds applied by the products, as well as their different snow depth biases (in case of reanalysis). Orsolini et al. (2019) already pointed out that the different thresholds applied by reanalysis datasets was one of the main limitations for inter-comparing them.*

*The sensitivity analysis also shows that changing 1 cm the station threshold leads to changes in the annual SCD bias of around 2-3 days. These changes are constant between products but vary between networks (ECCC = 2.8-4.3 days/cm, GHCN = 1.8-2.1 days/cm, RIHMI = 2.6-3.2 days/cm), due to the different snow*

*conditions in each station. Stations with more daily SD values close to the threshold are more affected by changes in the threshold."*

**Section 3.5: I suggest moving this into Section 4, because it is largely discussion material and does not present new analysis.**

**Answer**: We have moved it into a new Section 4, but we have kept the Conclusions in a separate section (new Section 5).

**Conclusions: The key result with respect to ERA5 is clearly stated on line 460: "In the reanalysis, data assimilation creates a trade-off between accuracy and stability." For applications like NWP, the instantaneous best estimate is the highest priority, but this of course does not ensure the temporal consistency required for climate monitoring. The key result for the NOAA CDR is communicated less clearly: "Overall, most of the trends/discontinuities observed are larger than the actual snow trends and the GCOS stability requirements, making these products inappropriate for climate applications without correction, particularly ERA5." I suggest re-phrasing this to provide an assessment more clearly focused on the NOAA CDR. This study provides a new line of evidence that autumn trends are very problematic in this dataset, but there are seasons and regions in which the product is suitable for climate analysis (e.g. MAM as shown in Figure 10b).**

**Answer**: We have rephrased NOAA CDR conclusions as follows:

*"NOAA CDR presents a positive artificial trend in SON and DJF. These results provide another line of evidence supporting the problematic fall trends in NOAA CDR and reveal that a similar trend appears in Europe of eastern North America during winter. Despite the numerous studies highlighting the inconsistency of NOAA CDR fall trends with in-situ measurements and with other datasets, some studies keep claiming a positive snow cover trend in fall based solely on NOAA CDR data (Cohen et al., 2021). Using NOAA CDR without correction in SON and DJF should be avoided. NOAA CDR could still be valid after correction, or in other regions and seasons (e.g., MAM) not affected by artificial trends"*

**Section 4 could also highlight that studies continue to claim there is a positive trend in autumn snow extent based solely on the NOAA CDR (https://doi.org/10.1126/science.abi9167) and do not acknowledge the literature which has identified problems with this dataset, so your study once again points out that this dataset is problematic in the autumn.**

**Answer**: We have rephrased NOAA CDR conclusions as shown in the previous comment.

**MINOR COMMENTS**

**Line 18: change 'snow cover decrease is aggravated' to 'snow cover decrease is coincident to decreasing snow depth…'**

**Answer**: Changed.

**Line 30: not clear what is meant by 'global circulation'.**

**Answer**: Global atmospheric circulation. We have clarified in the text.

**Paragraph 1 of the Introduction: The Stocker et al (2013) reference for snow trends and snow-albedo feedback is a little out of date. Updated SAF estimates are in the IPCC SROCC Chapter 3, and the Thackeray et al (2019) paper provides a fairly current review. (Meredith, M., M. Sommerkorn, S. Cassotta, C. Derksen, A. Ekaykin, A. Hollowed, G. Kofinas, A. Mackintosh, J. Melbourne-Thomas, M.M.C. Muelbert, G. Ottersen, H. Pritchard, and E.A.G. Schuur, 2019: Polar Regions. In: IPCC Special Report on the Ocean and Cryosphere in a Changing Climate [H.-O. Pörtner, D.C. Roberts, V. Masson-Delmotte, P. Zhai, M. Tignor, E. Poloczanska, K. Mintenbeck, A. Alegría, M. Nicolai, A. Okem, J. Petzold, B. Rama, N.M. Weyer (eds.)].)**

**Answer**: We have included the updated values provided in IPCC SROCC Chapter 3.

**Line 33: suggest changing to '…such as the Arctic and high elevations.'**

**Answer**: Changed.

**Line 33: "Notably, only 11 long-term stations are available in the Southern Hemisphere (SH)." Very interesting! Is there a reference for this statement?**

**Answer**: The statement was extracted from IPCC AR5:

*"Measurement challenges are particularly acute in the Southern Hemisphere (SH), where only about 11 long-duration in situ records continue to recent times: seven in the central Andes and four in southeast Australia."*

We have included the corresponding reference. Nevertheless, we have relaxed the sentence as follows:

*"Long-term snow measurements are particularly limited in the Southern Hemisphere (SH) (Stocker et al., 2013)."*

**Line 46: Is there a reference for the S-NPP VIIRS dataset, as is provided for the others in this list?**

**Answer**: We have added the reference to the product user manual.

**Line 50: This is a very minor point, but the most recent citation for the GlobSnow dataset (v3) is: Luojus, K., J. Pulliainen, M. Takala, J. Lemmetyinen, C. Mortimer, C. Derksen, L. Mudryk, M. Moisander, P. Venäläinen, M. Hiltunen, J. Ikonen, T. Smolander, J. Cohen, M. Salminen, K. Veijola, and J. Norberg. 2021. GlobSnow v3.0 Northern Hemisphere snow water equivalent dataset. Scientific Data. doi: 10.1038/s41597-021-00939-2.**

**Answer**: Updated.

**Line 87: "Since 2004, ERA5 also assimilates the IMS product but only over altitudes below 1500 m."
Could add a reference to the Orsolini et al (2017) paper here.**

**Answer**: We have added the reference.

**Line 101: "…but snow observations are not directly assimilated." This is a small point but make
clear that both the in situ snow depth and the IMS data are not assimilated into ERA5-land.**

**Answer**: We have clarified it as follows:

*"Neither in-situ snow depth measurements nor IMS data are directly assimilated by ERA5-Land."*

**Line 114: Some older citations could be added to provide readers with more background on the
CDR and IMS: Robinson, D., K. Dewey, and R. Heim. 1993. Global snow cover monitoring: an update.
Bulletin of the American Meteorological Society. 74(9): 1689-1696. Helfrich, S., D. McNamara, B.
Ramsay, T. Baldwin, and T. Kasheta. 2007. Enhancements to, and forthcoming developments in the
Interactive Multisensor Snow and Ice Mapping System (IMS). Hydrological Processes. 21: 1576-
1586.**

**Answer**: We have added both references.

**Line 120: remove 'around'**

**Answer**: Removed.

**Line 141: typo 'sires'**

**Answer**: Corrected.

**Line 223: "…stations are located either on peaks (Fig. 3b) or in the valley…" This wording is quite
specific. Perhaps just emphasize that elevation gradients around the stations create uncertainty?**

**Answer**: We have rephrased it as follows:

*"On mountainous regions, the spatial representativeness of the stations decreases due to the large
elevation gradients (Fig.3b)"*

**Line 267: I would not refer to the NOAA CDR as having a "retrieval algorithm" since it is analyst-
derived. How about "The positive bias is explained by changes in the analysis approach to produce
the snow charts, which since 1999…"**

**Answer**: We have rephrased it as follows:

*"The positive bias could be explained by changes in the analysis approach to produce the snow charts, which since 1999 considers a pixel snow-covered when only a 42% of the IMS pixels within the pixel were snow-covered."*

**Line 270: can you provide a reference to the NOAA CDR product manual?**

**Answer**: Yes, we have added the corresponding reference.

**Line 273: "…but a positive trend is observed since 1990 in fall and winter." Add a reference to Figure A2 here.**

**Answer**: Done.

**Line 285: The study of Mortimer et al (2020) focuses on the ERA5 discontinuity in 2004, not 1980. (please double check the other citations)**

**Answer**: The reviewer is right. Indeed, Mortimer et al (2020) reference was included in the discussion of the 2004 discontinuity.

Here we just wanted to state that ERA5 tends to have a positive bias in regions and periods when it does not assimilate station data (as reported by Mortimer et al (2020) above 1500 m). However, we have removed the reference from this section to avoid any misunderstanding.

**Line 324:Instead of Derksen, 2014, could cite Brown and Derksen (2013) here.**

**Answer**: We have changed the reference.

**Line 327: change 'algorithm' to 'analysts'**

**Answer**: Done.

**Line 392: "In regions such as Europe, spring SCD reductions add up to the decreasing SD, increasing, even more, the annual SCD trends." Awkward wording. I think the point is that in Europe both SD and SCD are decreasing, with the trend towards shallow snow depth amplifying the shorter SCD. In Russia, the snow cover season is shortening, despite positive SD trends in some areas, which means the spring melt signal driven by warming temperatures overrides any increase in snow accumulation during the winter.**

**Answer**: We have rephrased as follows:

*"In regions such as Europe, both SD and SCD are decreasing, with the trend towards shallow snow depth amplifying the shorter snow season. In Russia, spring SCD is also decreasing despite the positive trends in SD. This means that the spring melt driven by warming temperatures overrides any increase in snow accumulation during winter."*

**Line 453: I very much appreciate the comment that while multi-product ensembles are preferred for historical trend analysis, it is still important to quantify the performance of individual products over time.**

**Answer**: Thanks for your comment.

---

## Author Comment (AC2)

**Answers to reviewers: TC-2021-281**

**Temporal stability of long-term satellite and reanalysis products to monitor snow cover trends**

Ruben Urraca and Nadine Gobron

**REFEREE #2 – Alvaro Ayala**

Urraca and Gobron investigate the long-term temporal stability of snow-related variables produced by two global climate reanalysis products (ERA5 and ERA5-Land) for the period 1950-2020 (1980-2020 in the case of ERA5-Land) and the weekly Snow Cover Extent (SCE) charts produced by NOAA Climate Data Records (CDR) for the period 1966-2020. The authors compare these products against a set of 470 ground stations over the Northern Hemisphere. Temporal stability is investigated by calculating the bias in snow depth and snow cover duration of the products at each ground station. The authors found that the assimilation of new observations and satellite products improves the accuracy of snow variables of the reanalysis at the expense of introducing step discontinuities in the long-term time series, or, in the case of NOAA CDR, producing an artificial positive trend since 1990. Finally, the authors also use the ground stations data to update snow trends over the North Hemisphere.

I think that this a very good article, the research questions are clear and interesting for the scientific community and The Cryosphere. Results are well presented, and a clear message is provided in the Conclusions. The paper is a bit difficult to follow because it uses several data sets and methods, but overall, I think that the authors did a very good job in organizing the text. My recommendation is to accept the article with minor revisions. I have several short comments that could help for improving clarity.

**COMMENTS**

**1) Temporal stability. I have some suggestions to improve the use of this key term:**

**1.1) I think that adding the word "temporal" would make the title more informative "Temporal stability of long-term satellite…".**

**Answer**: We have added the word 'temporal' to the title:

*"Temporal stability of long-term satellite and reanalysis products for snow trend analysis"*

**1.2) Please provide a formal definition of temporal stability in the Introduction. Paragraph 5 could be a good option. In the glossary of GCOS (2016): "Stability may be thought of as the extent to which the uncertainty of measurement remains constant with time. In this publication, values in Annex A under "stability" refer to the maximum acceptable change in systematic error, usually per decade." The thresholds defined by GCOS could be written next to the chosen definition.**

**Answer**: We have included the definition given in GCOS (2016) in the introduction:

*"Stability is defined by GCOS as the extent to which the uncertainty of measurement remains constant with time (GCOS, 2016). GCOS stability requirements for snow cover are 10 mm/decade for SD and SWE, and*

*4%/decade for SCE. These requirements refer to the maximum acceptable change in systematic error per decade."*

**2) As the authors don't use data from Canada it might be good to comment about the limitations of the analyses of snow trends in the Northern Hemisphere.**

**Answer**: Following the reviewers' suggestions, we have added Canadian in-situ data to the study. We have processed all the stations available in the Canadian Historical Daily Snow Depth Database. Out of them, 57 passed our selection criteria for trend/stability analysis, and 34 were classified as representative for the validation of gridded datasets.

**SUGGESTED TECHNICAL CORRECTIONS**

**2: The acronym EO is not used again in the article. I would remove the parenthesis.**

**Answer**: We have removed the acronym.

**3: "Temporal stability is essential but…" Essential for what?**

**Answer**: We have rephrased the sentence as follows:

*"Monitoring snow cover to infer climate change impacts is now feasible using Earth Observation data together with reanalysis products (derived from earth system model and data assimilation). Temporal stability becomes essential when these products are used to monitor snow cover changes over time. The stability of satellite products can be altered when multiple sensors are combined into a single product, and due to the degradation and orbital drifts in each individual sensor."*

**5: "some longest satellite and reanalysis products" but NOAA CDR was not originally a satellite product, or yes? Maybe you can find a more general term than satellite?**

**Answer**: Before 1999, NOAA CDR was derived manually by trained scientists from optical satellite data. Since then, NOAA CDR assimilates the IMS satellite product. So, despite the manual processing of data, NOAA-CDR has always used satellite images as inputs.

**11: lack of direct data assimilation**

**and**

**11: at the expense of**

**Answer**: We have rephrased as follows:

*"By contrast, ERA5-Land is more stable because it does not assimilate directly snow observations, but this leads to a worse accuracy despite having a finer spatial resolution"*

**14-15: This sentence is a bit confusing. I would suggest using here the "trade-off" sentence of the conclusions.**

**Answer**: Rephrased as follows:

*"Reanalysis datasets face a trade-off between accuracy and stability when assimilating new data to improve their estimations."*

**25: What variable would be "snow-albedo feedback" with units W m-2 K-1? Can you be more specific?**

**Answer**: This is the definition of climate feedback given by IPCC:

*"Changes of the net energy budget at the top of atmosphere (TOA) in response to a change in the Global Surface Air Temperature (GSAT)"*

The IPPC AR reference is included at the end of the sentence for clarification.

**32: What do you mean by "changing vegetation"? Do you mean seasonal changes? Does it affect the spatial representativity of ground stations?**

**Answer**: The spatial representativeness of the stations, regarding snow cover variables, is reduced in stations surrounded by a heterogeneous land cover. We have rephrased as follows:

*"Ground stations provide the most accurate snow measurements, but their spatial representativeness is very limited in mountain regions or places with heterogeneous land cover"*

**34: What is the source for the 11 long-term stations in the Southern Hemisphere? In Chile and Argentina there are several snow stations with long-term data, although not with a very high frequency (Masiokas et al., 2006).**

**Answer**: The statement was extracted from IPCC AR5:

*"Measurement challenges are particularly acute in the Southern Hemisphere (SH), where only about 11 long-duration in situ records continue to recent times: seven in the central Andes and four in southeast Australia."*

We have included the corresponding reference. Nevertheless, we have relaxed the sentence as follows:

*"Long-term snow measurements are particularly limited in the Southern Hemisphere (SH) (Stocker et al., 2013)."*

**46: On the other hand, microwave-based…**

**Answer**: Done

**74-75: Please explain what data are used to update the trends.**

**Answer**: We have added the following clarification:

*"The study also updates the snow cover trends in the Northern Hemisphere from 1955 to 2015. Snow depth and snow cover trends are evaluated with in-situ data due to the discontinuities and trends found in gridded datasets. Snow cover extent could be only evaluated by inter-comparing the three gridded datasets."*

**98: "consistent with" wouldn't be more precise "derived from"?**

**Answer**: Rephrased as follows:

*"ERA5-Land is a replay of the land component of the ERA5 climate reanalysis, forced by meteorological fields from ERA5"*

**101-102: "but snow…" can be deleted as is a repetition from the previous paragraph.**

**Answer**: Done.

**141: series.**

**Answer**: Corrected.

**159: "The course products evaluated" this is a bit unclear as it seems that you are evaluating the reanalysis. Please replace by something like: "The coarse pixels correspond to that of ERA5 and ERA5-Land"**

**Answer**: We are indeed evaluating the quality of reanalysis products, so the interpretation is correct.

**160: Why did you choose 2015?**

**Answer**: We chose 2015 because it is the first full year provided by IMS 1km. We believe that one year of data is enough to evaluate the spatial representativeness of the stations, since we are covering all the snow cover patterns throughout the year.

**184: 50 or 5%?**

**Answer**: SCF = 50%.  We have modified Fig. 2 x-axis (previously it was in 0-10 scale) to be consistent with numbers in the text.

**185: in the middle of these values**

**Answer**: Done.

**226: affected by the station removal.**

**Answer**: Added.

**Answer**: Corrected

**252: RIHMI instead of RIHIMI. There might be more typos with this acronym, please revise.**

**Answer**: Thanks. We have revised the manuscript accordingly.

**257: lack of direct data assimilation**

**Answer**: Thanks. Added.

**274: In what figure can we see the positive trend?**

**Answer**: Fig A6 (fall) and Fig A7 (winter). We have specified the exact figures in the text.

**282: Delta bias was not defined as percentual. Please add the word "percentual" or similar. What would be the base for that percentage? Bias before?**

**Answer**: We have added the definition of both absolute and relative delta bias in the Methods section.

**361: …1950-2020 using data from the ground stations.**

**Answer**: Done.

**424: The acronym NWP has not been introduced**

**Answer**: We have defined it.

**Figure 2: Please add Snow Cover Fraction and Snow Depth in the caption.**

**Answer**: Thanks for the comment. We have added both.

**Figure 7: Why not showing the map of MAM in b?**

**Answer**: Because the number of significant trends in that season is smaller, and the few stations showing significant trends have diverging signs. Boxplots in Fig. A8 also evidence the lack of artificial trends globally in MAM.

We have clarified why we did not include this map in the figure caption.

**Figure 9: Please add in the caption that the trends are computed with the ground data.**

**Answer**: Done:

*"(a) Annual and (b) seasonal decadal trends in snow depth (SD) and snow cover duration (SCD) from 1955 to 2015 based on in-situ measurements"*

**Sometimes supplementary figures are named AX and sometimes SX.**

**Answer**: We have renamed all supplementary figures consistently.

**Figure A2: Is panel b RIHMI-NOAA CDR correct? It seems that there are very few valid observations. These data the same as those used in Figure 7a, aren't they?**

**Answer**: Yes. Panel b (SCD bias in winter, new Fig. A7) seems to be empty because the SCD bias is zero or close to zero in most RHIMI stations during winter. This is because most RIHMI stations are covered by snow during the whole winter period.

**Table 2: According to line 170, the units of SSE and SSB should be days/year.**

**Answer**: Yes. We have updated it accordingly.

---

## Author Response (AR2)

**Answers to editor: TC-2021-281**

**Temporal stability of long-term satellite and reanalysis products to monitor snow cover trends**

Ruben Urraca and Nadine Gobron

**PUBLIC EDITOR COMMENTS**

Dear authors,

Thank you very much for the thorough revisions in response to the two reviewers' comments. Some of those comments were major ones, and I appreciate the new analyses you carried out in response to those major issues. I appreciate in particular the inclusion of all new Canadian stations, which increased the number of stations analysed from 470 to 527; and the efforts to obtain and remove from your validation the list of stations that are assimilated in ERA. I think indeed this was an excellent comment by reviewer 1 and commend you for the effort made to address it. I wonder if this list could be made public as an appendix to the paper – upon clarification with the ERA team. I also think that the sensitivity analysis included in response to the comment on how to identify discontinuity has strengthened the paper.

Given that the paper underwent major revisions, I am sending it back to the reviewers. I feel that some more explanations (and possibly analysis) are needed to address reviewer's 1 comment on why the authors decided to include the stations which failed the spatial representative test in the trend analysis, as those trends identified will be very local. I will first leave this to the reviewer to evaluate.

Once again, thanks for the thorough revision.

All my best,

**Answer**: Regarding the concern of how low spatially representative stations affect the trend analysis:

Spatial representativeness is critical when comparing satellite pixels to in-situ measurements (point-to-pixel comparison), as it introduces a spatial mismatch between satellite values and in-situ values. Therefore, stations where this mismatch error is significant (due to a high spatial variability and/or a coarse resolution of satellite pixel) need to be discarded or corrected using an up-scaling procedure.

However, stations with low spatial representativeness can still be useful for other applications such as trend analysis. This has been discussed by previous studies such as *Schwartz et al 2017.* A good example are stations on the coast (Norwegian coast, Canadian coast or Eastern Russia). All of them are excluded for the point-to-pixel comparison because the satellite pixel includes the sea. However, the trends observed are coastal locations are still valid from a climate perspective.

Following the editor comment, we have compared the decadal trends obtained in the group of low spatially representative stations against those obtained in the group of spatially representative stations (Figure 1). We do not observe any inconsistency between both groups.

We have also clarified in the Methods sections the limitations of using in-situ data for evaluating global trends:

*"Note that the density of stations was too low for a complete analysis of NH snow cover trends. Even in regions with good coverage, the heterogeneous density of the stations as well as their different spatial representativeness also prevent the calculation of spatial representative trends. However, our main goal*

*was to estimate the trend magnitude to evaluate the significance of the artificial trends and discontinuities introduced by each product."*

References

M. Schwarz, D. Folini, M. Z. Hakuba, M. Wild. Spatial Representativeness of Surface-Measured Variations of Downward Solar Radiation. JGR Atmospheres 2017.https://doi.org/10.1002/2017JD027261

[Figure]

**Fig 1** Decadal trends in snow depth (SD) and snow cover duration (SCD) from 1955 to 2015 based on in-situ measurements, splitting between spatially representative and non-spatially representative stations.

**NON-PUBLIC EDITOR COMMENTS**

Dear authors,

Thanks again for the thorough revisions, which have improved the paper considerably. Just a minor note here: I would have appreciated in few responses a short summary of how the results have changed as a result of the new analysis carried out, e.g. does inclusion of the new canadian stations or removal of almost half of the validation stations change your major results or conclusions?

**Answer**:

The study is composed of three main sections: (1) stability analysis of gridded products, (2) accuracy analysis of gridded products, and (3) snow trend estimation based on in-situ data. The changes mentioned by the editor affected the different sections as follows:

- Addition of Canadian stations.

  1. Stability analysis: The new Canadian stations corroborated that the ERA5 step discontinuity observed between 1977-1980 exists at global scale. Besides, the analysis of the stations assimilated by ERA5 over Canada corroborated that this discontinuity was most likely due to starting assimilating in-situ snow observations during those years (1977-1980).

  2. Accuracy analysis: Most Canadian stations were low spatially representative, so their influence in this section was negligible.

  3. Trend estimation: This is the section where the addition of Canadian data was most valuable, as now we have a better coverage of NH regions with seasonal snow. Negative trends in both snow depth and snow cover duration were obtained in most of the Canadian stations analyzed.

- Identification of stations assimilated by ERA5.

  1. Stability analysis: We used the list of stations assimilated by ERA5 to identify the cause of the step discontinuities observed in the temporal evolution of ERA5 bias (Figure 6). We could confirm that both the global discontinuity in 1977-1980 and the Asian discontinuity in 1991-1992 were caused by the addition of new snow in-situ measurements to the ERA5 model in those regions during those years.

  2. Accuracy analysis: The number of stations available was significantly reduced in order to guarantee the independence of our validation set. As expected, ERA5 performance metrics (bias, RMSE) slightly worsen after removing the stations assimilated by ERA5.

  3. Trend estimation: this part of the study was not affected, as decadal trends were calculated directly from the in-situ measurements.

Could you double check with the ERA team whether the list of stations assimilated can be made public in an appendix? I think it would make a great service to the community.

**Answer**: We have not been able to confirm with C3S/ECMWF whether the exact list of stations assimilated (and the exact year of inclusion) can be included in the appendix. They suggested that the full list of stations

assimilated (not only snow but other variables) may be included in ERA5 documentation page in upcoming releases.

If this situation changes during the review process of the paper, we will add the corresponding table.

**I would also appreciate some more explanations as to why you chose to evaluate only the ERA and ERA5 snow products in the introduction, compared to other available re-analysis. The Japanese reanalysis for instance cover almost the same period as ERA (from 1953 onwards). I feel it would lend strength to your manuscript.**

**Answer**: The goal of the study is to analyze the different stability issues faced by satellite products, global reanalysis, and land reanalysis to monitor long term snow trends. Therefore, we selected one product of each type (the ones with the longest coverage and typically also the ones most used): NOAA CDR, ERA5 and ERA5-Land. We have summarized other long-term products available in the introduction, either reanalysis ones (JRA-55, MERRA-2) or satellite ones (JAXA GHRM5, ESA Globsnow, ESA CCI SWE).

At this point, adding a new gridded product would require a huge amount of work. Our study is based on daily snow depth/cover values, but reanalysis typically provides the data as monthly or hourly averages. In the case of JRA-55, we would need to download and process 3-hourly snow depth data for almost 70 years, which will require a lot of computational time.

**Otherwise I look forward to the reviewers' second evaluation and hope the paper can be published soon. It contains some very important results that will be very useful to the community.**

**All my best,**

**Francesca**

---

## Author Response (AR3)

**Answers to reviewers: TC-2021-281**

**Temporal stability of long-term satellite and reanalysis products to monitor snow cover trends**

Ruben Urraca and Nadine Gobron

**REVIEWER #1**

Thanks very much to the authors for their thorough revisions to the manuscript. In particular, I appreciate the effort to add data from Canada, which means there is now better spatial coverage of the surface snow observations. I have a small number of final comments for the authors to consider:

1. The methodology now explicitly describes the identification of stations that were assimilated into ERA5, and how these were treated within the analysis (Line 165: "These stations were kept for the stability analysis, since their addition to ERA5 may explain some of the discontinuities observed, but were removed from the accuracy analysis to guarantee the independence of the validation set.").

In line 264 it states that 387 stations were used for validation; how many of these stations are part of the 235 stations assimilated into ERA5? Is it possible to use different symbol shapes in Figure 3a to indicate which stations are assimilated?

**Answer**: The total number of stations used in the study is 527. For the validation (accuracy analysis), we used spatially representative stations (387 out of 527) that are not assimilated into ERA5 (214 out of 527). The number of stations meeting both conditions is 152 out of 527. We have clarified this in the manuscript.

*"Only spatially representative stations that are not assimilated by ERA5 are used (152 out of 527)."*

As suggested by the reviewer, we have added different symbol shapes (circles and triangles) in Fig 3a to indicate which stations are assimilated into ERA5.

2. Justification for the use of the 2.5 cm snow depth threshold for determining snow cover (and subsequently snow cover duration) is now more clearly described. As noted in my original review, I appreciate the effort taken to quantify the spatial representativeness of the point measurements.

**Answer**: Many thanks for this positive comment.

3. As noted in my review of the original manuscript, the effort to address the fall trends in the NOAA CDR is important. In Section 5, the key findings are clearly stated (lines 517-522). The original presentation of these findings in Section 3.2.2 is, however, written less clearly. I think this is because the spurious nature of the NOAA CDR fall trends is illustrated by a positive trend in the bias (as opposed to Figure 11 which shows the false trend in actual snow extent), and the wording is sometimes ambiguous as to whether the trend in extent is positive or the trend in bias is positive. I suggest revising Section 3.2.2 for clarity, so that it more clearly links to the conclusions in Section 5.

**Answer**: Thanks for the comment. We have revised this section clarifying if we referred to 'the trend in the bias of snow cover duration' or to 'the trend in snow cover extent'

4. In general, the manuscript is clearly written, but a final editorial review for grammar would be helpful.

**Answer**: We have performed a final editorial review.

**Editorial**

**Abstract, line 14: suggest changing "…with the increasing number of satellite data used." to "…with changes to the available satellite data."**

**Answer**: Done.

**Abstract line 21: suggest changing to "…while in drier regions such as Russia earlier snow melt occurs despite increased maximum seasonal snow depth."**

**Answer**: Done.

**Line 69: Since the first version of this manuscript was submitted, a new manuscript has come out which shows the SMMR to SSM/I transition creates a temporal inhomogeneity in the Snow CCI dataset. I leave it up to the authors to decide whether there is value to add a citation.**

**Mortimer, C., L. Mudryk, C. Derksen, M. Brady, K. Luojus, P. Venäläinen, M. Moisander, J. Lemmetyinen, M. Takala, C. Tanis, and J. Pulliainen. 2022. Benchmarking algorithm changes to the Snow CCI+ snow water equivalent product. Remote Sensing of Environment. DOI: 10.1016/j.rse.2022.112988.**

**Answer**: Thanks for the update. We have included the new reference.

**Line 306: consider changing the wording here to "The positive trend in fall has been previously reported as problematic in several studies."**

**Answer**: Changed.

**Figure 6: One too many C's in the ECCC label.**

**Answer**: We have corrected this typo in all the figures.

**REVIEWER #2**

Urraca and Gobron investigate the long-term temporal stability of snow-related variables produced by two global climate reanalysis products (ERA5 and ERA5-Land) for the period 1950-2020 and the weekly Snow Cover Extent (SCE) charts produced by NOAA Climate Data Records (CDR) for the period 1966-2020.

This article is a valuable contribution as I believe that its results are interesting for the scientific community and The Cryosphere. I only have some minor comments that could help to improve some details regarding presentation and take-home messages.

**COMMENTS**

I would add to the manuscript the answer that you give the Editor about why you did not include in your article other products such as the Japanese reanalysis. I think that you provide a reasonable argument (one product of each type).

**Answer**: We have added the following clarification in the introduction:

*"The goal of the study is to analyze the temporal stability of snow-related variables form satellite products, global reanalysis, and land reanalysis. We selected one product of each type: NOAA CDR (1966-present), ERA5 (1950-present), and ERA5-Land (1950-present), respectively. They provide the longest temporal coverage in each group."*

**SUGGESTED TEXT CORRECTIONS**

**3: Remove the parenthesis**

**Answer**: Done.

**3: While the temporal stability of satellite products …, the stability of reanalysis datasets…**

**Answer**: Done.

**7: Some of the longest**

**Answer**: Done.

**30: a strong snow-albedo feedback**

**Answer**: Done.

**150: the 1980s**

**Answer**: Done.

**151, 153: the 1990s**

**Answer**: Done.

**157: due to the low number of available ECCC stations…**

**Answer**: Done.

**158: at least 90%**

**Answer**: Done.

**513: In the reanalysis, the incorporation of new data to the data assimilation creates a trade-off between accuracy and stability**

**Answer**: Done.

**520: e.g. Cohen et al.**

**Answer**: Done.

**FIGURES**

**Figure 3: (b) coastal station, (c) mountain station**

**Answer**: Done.

**Figure 3: In what panel are the stations in grey?**

**Answer**: Pannel (a). For clarification, we have moved the description of stations in grey just after the panel description.

**Figure 4: What is the cause for the last big boxplot in ERA5-ECCCC panel?**

**Answer**: Despite having enough number of stations, the ECCC stations available in 2015 (last boxplot) were not as homogeneously distributed over Canada (as during the previous years). Thus, we have removed that year from all plots that aggregate all ECCC stations to one annual metric (like boxplots).

**Figure 7 (caption): 2003-04 discontinuities**

**Answer**: Corrected.